# COUNTERFACTUAL REALIZABILITY

**Arvind Raghavan** and **Elias Bareinboim**

Causal Artificial Intelligence Lab
Columbia University
{ar, eb}@cs.columbia.edu

## ABSTRACT

It is commonly believed that, in a real-world environment, samples can only be drawn from observational and interventional distributions, corresponding to Layers 1 and 2 of the *Pearl Causal Hierarchy*. Layer 3, representing counterfactual distributions, is believed to be inaccessible by definition. However, Bareinboim, Forney, and Pearl (2015) introduced a procedure that allows an agent to sample directly from a counterfactual distribution, leaving open the question of what other counterfactual quantities can be estimated directly via physical experimentation. We resolve this by introducing a formal definition of *realizability*, the ability to draw samples from a distribution, and then developing a complete algorithm to determine whether an arbitrary counterfactual distribution is realizable given fundamental physical constraints, such as the inability to go back in time and subject the same unit to a different experimental condition. We illustrate the implications of this new framework for counterfactual data collection using motivating examples from causal fairness and causal reinforcement learning. While the baseline approach in these motivating settings typically follows an interventional or observational strategy, we show that a counterfactual strategy provably dominates both.

## 1 INTRODUCTION

The *Pearl Causal Hierarchy*, or PCH, is an important recent milestone in our understanding of causality (Pearl & Mackenzie, 2018; Bareinboim et al., 2022). The three layers of the PCH represent the distinct regimes of *seeing*, *doing*, and *imagining*, with regard to an environment. Consider an environment involving a decision variable $X$ and an outcome $Y$. Layer 1 ($\mathcal{L}_1$) represents *observational* distributions, such as $P(Y \mid x)$. Layer 2 ($\mathcal{L}_2$) represents *interventional* distributions, such as $P(Y; do(x))$, using the $do()$ operator. Layer 3 ($\mathcal{L}_3$) represents *counterfactual* distributions dealing with conflicting realities, such as $P(Y_x \mid x', y')$: the distribution of $Y$ had $X$ been fixed as $x$, given that $X, Y$ were in fact naturally observed to be $x', y'$. Higher layers subsume lower ones, but are underdetermined by them (Ibeling & Icard, 2020; Bareinboim et al., 2022).

Reasoning about $\mathcal{L}_3$-quantities plays a vital role in personalized decision-making (Mueller & Pearl, 2023), analysing a causal effect into direct and indirect pathways (Pearl, 2001; Rubin, 2004), and constructing explanations for decisions, among other topics, in applications such as healthcare (Mueller & Pearl, 2024), economics (Li & Pearl, 2019), epidemiology (Robins & Greenland, 1992) etc. Suppose an economist were interested in estimating $P(y_x \mid x')$, an important $\mathcal{L}_3$-quantity called the effect of the treatment on the treated, or ETT (Heckman & Robb Jr., 1985; 1986). One approach to computing such quantities is through *identification* (Pearl, 2000, §3.2.4): leveraging causal knowledge about the environment, typically a causal graph or parametric assumptions, to infer the higher-layer quantity using lower-layer data. This approach fails when the quantity is nonidentifiable, e.g. ETT in the general setting (Shpitser & Pearl, 2009; Correa et al., 2021).

However, another approach uses physical experimentation to attempt to directly draw samples from the relevant distribution, $P(Y_x, X)$ in the case of ETT, and then uses statistical methods to estimate $P(Y_x = y, X = x')$. This approach is only possible if there is some sequence of physical actions by which an agent can measure these random variables simultaneously for a single unit. It is generally believed to be feasible to draw samples only from $\mathcal{L}_1$- and $\mathcal{L}_2$-distributions, the latter by interventions like randomized controlled trials (RCT), à la Fisher (Fisher, 1935), and the former by simply observing the natural behaviour of the system. $\mathcal{L}_3$-distributions like $P(Y_x, X)$ are deemed

non-realizable in general, as the potential response $Y_x$ and natural decision $X$ belong to different "worlds". Once a unit naturally adopts decision $X = x'$, $Y_x$ cannot be evaluated in the $do(x)$ regime for the same unit.[1] However, Bareinboim, Forney & Pearl have shown it is feasible to draw samples from the ETT distribution $P(Y_x, X)$ through a *counterfactual randomization* procedure (Bareinboim et al., 2015; Forney et al., 2017). This leaves open the possibility that other $\mathcal{L}_3$-distributions, say perhaps $P(Y_x, X, Y)$, are also realizable through clever experimental setups, allowing one to estimate important quantities like the probability of sufficiency, $P(y_x \mid y', x')$ (Pearl, 1999).

This brings us to the central question motivating this work: *from which $\mathcal{L}_3$-distributions is it possible to draw samples given fundamental physical constraints like the inability to travel back in time and subject the original unit to a different experimental condition?* We resolve this open question with a rigorous formal treatment of the *realizability of an $\mathcal{L}_3$-distribution* (Def. 3.4).

Our main contributions in this work are as follows:

- In Sec. 2 we introduce a physical procedure called *counterfactual randomization* (Def. 2.3) by which an agent can gather counterfactual data, subsuming previous similar notions.

- In Sec. 3 we develop the **CTF-REALIZE** algorithm (Algo. 1) to determine whether an $\mathcal{L}_3$-distribution is physically realizable. We prove the algorithm is complete (Thm. 3.5), and derive important corollaries characterizing realizable distributions (Cors. 3.7,3.8). For instance, we show that our main result generalizes an influential notion in the causal inference literature, known as the *fundamental problem of causal inference* (Holland, 1986).

- In Sec. 4 we discuss important practical implications of counterfactual realizability. The traditional route of computing $\mathcal{L}_3$-quantities through identification often fails. Our work suggests opportunities for novel experiment-design ideas to directly estimate these quantities, as illustrated through Examples 1,2 and 3. More concretely,

  - In Sec. 4.1, we describe an application in causal fairness, where the naive approach of constraining a classifier using an interventional ($\mathcal{L}_2$) fairness metric fails to prevent disparities in outcomes across groups, but where a counterfactual ($\mathcal{L}_3$) approach works.

  - In Sec. 4.2, we show how counterfactual randomization can be used to improve RL algorithms. The baseline approach in a multi-arm bandit setting is to use allocation procedures (e.g., UCB, EXP3, Thompson Sampling) to discover which arm $x$ optimizes the expected outcome $\mathbb{E}[Y; do(x)]$, which is an interventional ($\mathcal{L}_2$) strategy (Sutton & Barto, 1998; Lattimore & Szepesvári, 2020). It turns out there are provably superior strategies (w.r.t expected outcome) based on directly optimizing counterfactual ($\mathcal{L}_3$) objectives, as we demonstrate in Example 3. We prove optimality of our proposed strategy in a bandit setting with a generic causal template (Thm. 4.2, Cor. 4.3 in Raghavan & Bareinboim (2025)).

Proofs and experiment details are in the full technical report (Raghavan & Bareinboim, 2025).

**Preliminaries.** We denote variables by capital letters, $X$, and values by small letters, $x$. Bold letters, $\mathbf{X}$, are sets of variables and $\mathbf{x}$ sets of values. $P(\mathbf{x})$ is shorthand for $P(\mathbf{X} = \mathbf{x})$. $\mathbb{1}[.]$ is the indicator function. We use *Structural Causal Models* (SCM) to describe the generative process for a system of interest (Bareinboim et al., 2022, Def. 1)(Pearl, 2000). An SCM $\mathcal{M}$ is a tuple $\langle \mathbf{V}, \mathbf{U}, \mathcal{F}, P(\mathbf{u}) \rangle$. $\mathbf{V}$ is the set of observable variables. $\mathbf{U}$ is the set of unobservable variables exogenous to the system, distributed according to $P^{\mathcal{M}}(\mathbf{U})$. $\mathcal{F} = \{f_V\}$ is a set of functions s.t. each $f_V$ causally generates the value of $V \in \mathbf{V}$ as $V \leftarrow f_V(\mathbf{U}_V, \mathbf{Pa}_V)$, where $\mathbf{U}_V \subseteq \mathbf{U}$ and $\mathbf{Pa}_V \in \mathbf{V} \setminus V$. Each $\mathcal{M}$ induces a *causal diagram* $\mathcal{G}$ (Bareinboim et al., 2022, Def. 13), which is a graph containing a vertex for each $V \in \mathbf{V}$, a directed edge from each node in $\mathbf{Pa}_V$ to $V$, and a bidirected edge between $V, V'$ if $\mathbf{U}_V, \mathbf{U}_{V'}$ are not independent. Given a graph $\mathcal{G}$, $\mathcal{G}_{\overline{\mathbf{X}}\underline{\mathbf{W}}}$ is the result of removing edges coming into variables in $\mathbf{X}$, and edges coming out of $\mathbf{W}$. We use standard terminology like parents, descendants of a node (see App. A). Our treatment is limited to *recursive* SCMs, which implies acyclic diagrams, with finite discrete domains over $\mathbf{V}$. The $do(\mathbf{x})$ operator indexes a sub-model $\mathcal{M}_\mathbf{x}$ where the functions generating variables $\mathbf{X}$ are replaced with constant values $\mathbf{x}$. A variable $Y \notin \mathbf{X}$ evaluated in this regime is called a *potential response*, denoted

---

[1] E.g., "The problem with counterfactuals like $[P(Y_x \mid x')]$ is [that] ... we simply cannot perform an experiment where the same person is both given and not given treatment." (Shpitser & Pearl, 2007) Also, "By definition, one can never observe [counterfactuals], nor assess empirically the validity of any modeling assumptions made about them..." (Dawid, 2000)

$Y_{\mathbf{x}}$. $(\mathbf{W}_\star = \mathbf{w})$ denotes an arbitrary counterfactual event, e.g. $(Y_x = y \wedge Y_{x'} = y' \wedge X = x'')$. The probability of such an event is given by the $\mathcal{L}_3$-valuation (Bareinboim et al., 2022, Def. 7):

$$P^{\mathcal{M}}(\mathbf{W}_\star = \mathbf{w}) = \sum_{\mathbf{u}} \left( \prod_{W_{\mathbf{t}} \in \mathbf{W}_\star} \mathbb{1}[W_{\mathbf{t}}(\mathbf{u}) = w] \right) P^{\mathcal{M}}(\mathbf{u}), \text{ with } w \text{ taken from } \mathbf{w}.$$

## 2 DATA-COLLECTION PROCEDURES

In this section, we define a procedure, *counterfactual randomization*, that extends the scope of traditional *Fisherian* experimentation (discussed below). Consider a system of interest modeled by unknown SCM $\mathcal{M}$. Interventions and counterfactual events are typically defined in terms of *symbolic* operations on $\mathcal{M}$. To conceptually separate this from the *physical* constraints experienced by an agent (natural or artificial), we define the following physical actions that an agent can perform in the system. These are simply the physical counterparts to symbolic procedures.

We call each discrete episode of the system's behaviour a *unit*. Examples of units are patients in a clinical trial, neighbourhoods in a social science experiment, rounds played on a slot machine etc. We index units w.l.o.g. by $i = 1, 2, 3...$, which constitute a target population in the system.

**Definition 2.1** (Physical actions). (1) $\text{SELECT}^{(i)}$: randomly choosing, without replacement, a unit $i$ from the target population, to observe in the system; (2) $\text{READ}(V)^{(i)}$: measuring the realized feature $V^{(i)}$ of unit $i$, produced by a causal mechanism $f_V \in \mathcal{F}$ operating on $i$; (3) $\text{RAND}(X)^{(i)}$: erasing and replacing $i$'s natural mechanism $f_X$ for a decision variable $X$ with an enforced value drawn from a randomizing device having support over $\text{Domain}(X)$. ∎

$\text{READ}(V)^{(i)} = v$ and $\text{RAND}(X)^{(i)} = x$ are also overloaded to refer to the values read and enforced, respectively. $\text{RAND}(X)^{(i)}$ is the standard Fisherian randomization of a decision variable $X$, corresponding to the symbolic procedure of a *stochastic* intervention on $X$ (Correa & Bareinboim, 2020).[2] As $\text{RAND}(X)^{(i)}$ erases the unit $i$'s natural decision, $\text{READ}(X)^{(i)}$ will yield the value randomly assigned to unit $i$. The discovery of this procedure marked an important achievement in the history of science and experiment-design (Fisher, 1925; 1935). Since the use of a randomizing device eliminates by design any confounding between the assigned decision and the unit's latent attributes $\mathbf{U}^{(i)}$, it allows researchers to estimate causal effects.

It is evident that the actions in Def. 2.1 are sufficient for an agent to physically draw samples from any $\mathcal{L}_1$- or $\mathcal{L}_2$-distribution, as discussed in App. C.1. Until recently, it was generally presumed these were the only physical actions possible on units in a system. However, we discuss some important extensions of experimental capabilities next.

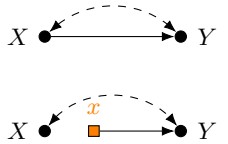

Figure 1: (Top) Causal diagram with decision variable $X$; (Bottom) Procedure of randomizing the actual decision without erasing the unit's natural decision.

**Counterfactual data-collection procedures.** In an early work from the causal reinforcement learning literature, Bareinboim, Forney & Pearl describe an experimental setting in which it is possible to both randomize a unit's actual decision, and also record the natural decision the unit *would have normally* taken (Bareinboim et al., 2015; Forney et al., 2017). Subsequently, this procedure has been used to establish benchmarks in counterfactual decision making (Zhang & Bareinboim, 2022). These settings involve an agent introspecting to gauge their natural choice, or otherwise revealing their natural choice by some indication, e.g. physical gestures prior to decision-time. Importantly, this form of randomization does not erase the unit's natural choice of decision variable $X$, as schematically illustrated in Fig. 1.

Building on the idea, we formalize this into a more general extension of the agent's capabilities: the ability to intervene on a variable $X$'s value *as perceived* by its causal children. To illustrate this, consider the $\mathcal{L}_3$-quantity known as natural direct effect, or NDE, which is used in mediation analysis to measure the effect of $X$ on $Y$ via a "direct" path, as opposed to an "indirect" path via a mediator $Z$ (Pearl, 2001) – highly relevant in several fields, as discussed in Sec. 1. The NDE is generally considered as identifiable from experimental data only under certain conditions (Pearl, 2001; Correa et al., 2021). The following example details an experiment design where it is possible to compute the NDE even when these identification conditions are not met, by randomizing the *perception* of $X$.

[2]If the device used for enforcing the value of $X$ is a constant function, this action simply becomes $\text{WRITE}(X : x)^{(i)}$, corresponding to the atomic intervention $do(x)$. See Preliminaries in Sec. 1.

**Example 1 (Mediation analysis).** A computer vision company's tool is being evaluated for an automated speeding ticket system that uses footage from traffic cameras. But the government's audit team has a concern: it is possible the model is trained on footage with a strong correlation between the color of the car and speeding (perhaps due to color preference of different socioeconomic neighbourhoods), and unfairly penalizes certain car colors.

This amounts to a hypothesis that $X$ (car's color) affects $Y$ (AI decision to issue a ticket) via a direct path as opposed to the indirect path via $Z$ (speeding). The indirect path describes the causal effects of, say, how pedestrians and other drivers react to a red car and affect its speeding. This hypothesis is true iff NDE is measured to be non-0, where NDE is defined as the following expression: $\text{NDE}_{x,x'}(y) = P(y_{x'Z_x}) - P(y_x)$ (Pearl, 2001). The second term, $P(y_x)$, can be estimated from a Fisherian randomization of $X$ (say, an experiment recruiting drivers and assigning them random cars). Inconveniently, the first term, $P(y_{x'Z_x})$, is nonidentifiable for Fig. 2(a), even using RCT data. So it is unclear how to make progress with this hypothesis test.

However, the audit team recognizes there exists a special mediator, viz. the features $W$ in the video which reveal the car's color to the model (say, RGB values of pixels in the video frames). They use standard video-editing tools to randomly swap the color of the car in the footage. By randomly assigning a particular car $W \leftarrow$ red, they are able to affect the mechanism $f_Y$'s *perception* of $X$:

$$
\begin{aligned}
& P(Y_{W=\text{red}} \mid X = \text{blue}) && \text{est. from } \mathcal{L}_2 \text{ data} && (1) \\
=& P(Y_{W=\text{red},Z} \mid X = \text{blue}) && Z : \text{natural value} && (2) \\
=& P(Y_{W=\text{red},Z_{X=\text{blue}}} \mid X = \text{blue}) && \text{consistency property} && (3) \\
=& P(Y_{X=\text{red},Z_{X=\text{blue}}} \mid X = \text{blue}) && \text{Def. 2.2, } X \equiv W && (4) \\
=& P(Y_{X=\text{red},Z_{X=\text{blue}}}) && \text{d-separation} && (5)
\end{aligned}
$$

Figure 2: (a) "Expanded" diagram for Example 1, where $W$ is *counterfactual mediator* for $X$; (b) Randomizing the value of $X$ as perceived by $Y$.

Eq. 4 is justified because $W$ controls $Y$'s perception of $X$ given a fixed $z$ (formalized in Lemma E.4 in the full technical report here). Thus, they are able to directly sample from the $\mathcal{L}_3$-distribution $P(Y_{x'Z_x}, X)$ via a physical procedure, and use identification rules to obtain $P(y_{x'Z_x})$. Using the formula for NDE, they can evaluate whether a car's color has a direct effect on the odds of getting a speeding ticket. ∎

Here, one is able to randomize $X$ as perceived by one of its children, by leveraging the variable $W$ (RGB values) that fully encodes information about $X$ (color) and mediates its effect on $Y$. We capture this intuition with the following (informal) definition.

**Definition 2.2 (Counterfactual mediator (informal)).** We call $W$ a *counterfactual mediator* of $X$ w.r.t $Y \in Ch(X)$ if the value of $X$ can be retrieved from $W$ by the mechanism generating $Y$. ∎

Other examples of interventions on perceived attributes via counterfactual mediators include changing details on a job application (name, pronouns, keywords) to simulate a perceived alternate demographic identity (Bertrand & Mullainathan, 2003), or editing specific portions of text input to a language model (Feder et al., 2022). Randomizing perception has been discussed in Pearl et al. (2016, §4.4.4). For a detailed discussion of the causal semantics of intervening on perceptions, and the related literature, see (Plecko & Bareinboim, 2024, App. D.1). We also provide a rigorous treatment in App. E of the full technical report, including a formal Def. E.2 of a counterfactual mediator.

This important extension to experimental capabilities is captured in the following definition of a new physical action that an agent might be able to perform in an environment.

**Definition 2.3 (Counterfactual (ctf-) randomization).** CTF-RAND$(X \to \mathbf{C})^{(i)}$: fixing the value of $X$ *as an input to* the mechanisms generating $\mathbf{C} \subseteq Ch(X)_\mathcal{G}$ using a randomizing device having support over Domain$(X)$, for unit $i$, given causal diagram $\mathcal{G}$. ∎

The key differences between the Fisherian RAND$(X)^{(i)}$ and CTF-RAND$(X \to \mathbf{C})^{(i)}$ are (1) CTF-RAND does not erase the unit $i$'s natural decision $X^{(i)}$; and (2) while RAND affects all children of $X$, CTF-RAND does not affect $Ch(X) \setminus \mathbf{C}$. CTF-RAND can only be enacted under certain structural conditions, viz., either in environments which permit the measurement of a unit's natural decision while simultaneously randomizing the actual decision (Bareinboim et al., 2015), or where counterfactual mediators can be used to alter $X$ as perceived by a subset of children. Whether the agent is indeed able to perform this action thus depends on the specific experimental setting.

Note: Def. 2.3 implies that it is possible to physically perform multiple randomizations involving the same variable $X$ on a single unit $i$, with each intervention affecting a different subset of children. Further, CTF-RAND may only be performed w.r.t a graphical child variable; it is not possible to bypass a child and directly affect a descendant's perception of $X$.

## 3 COUNTERFACTUAL REALIZABILITY

Given the possibility of performing ctf-randomization (Def. 2.3), we are interested in knowing which $\mathcal{L}_3$-distributions can be accessed directly by experimentation. In this section, we discuss the constraints imposed by nature on an agent. We then formally define *realizability* and develop a complete algorithm to determine whether an $\mathcal{L}_3$-distribution is realizable.

The most basic constraints experienced by the agent (natural or artificial) are physical. Each mechanism $f_V \in \mathcal{F}$ represents some physical process that transforms a unit $i$ according to the laws of nature. For instance, taking a drug, $X$, produces a side effect in the patient, $Y$, by a biochemical reaction $f_Y(X, U_Y)$, which depends on the drug and the patient's latent health condition, $U_Y$. Once patient $i$ has been subjected to mechanism $f_Y$ under $X = x$, there appears to be no way to go back in time and subject the same patient to mechanism $f_Y$ under $X = x'$. Even if technologically feasible to reverse the process (e.g., by taking an antidote to the drug), the latent factors $\mathbf{U} = \mathbf{u}$ might have changed after the experiment (e.g., the patient could have developed tolerance to the drug). Repeating the experiment on this patient is tantamount to testing a *new* unit with unknown latent features $\mathbf{U} = \mathbf{u}'$.[3] This observation is made more formal through the following assumption.

**Assumption 3.1** (Fundamental constraint of experimentation (FCE)). A unit $i$ in the target population can physically undergo a causal mechanism $f_V \in \mathcal{F}$ at most once. ∎

*Remark* 3.2. The FCE assumption entails that a unit $i$ can only be submitted to a particular mechanism $f_V(\mathbf{Pa}_V, \mathbf{U}_V)$ under a single set of experimental conditions, received as input to $f_V$. By implication, the physical actions in Defs. 2.1, 2.3 can only be performed at most once per unit $i$. ∎

Once unit $i$ has been subjected to $f_V$, it is not possible to re-run $f_V$ with differently fixed inputs. READ$(V)^{(i)}$ thus only yields one value for $i$. Although ctf-randomization permits multiple interventions involving the same variable $X$, each such intervention can only be performed once, since it impacts different child mechanisms that can each only occur once for unit $i$. We also assume that the agent can only perform the physical actions in Defs. 2.1, 2.3, up to isomorphism.

**Definition 3.3** (I.i.d sample). Given an $\mathcal{L}_3$-distribution $Q = P(\mathbf{W}_\star)$ and a sequence of physical actions $\mathcal{A}^{(i)}$ performed on unit $i$ in an environment modeled by SCM $\mathcal{M}$, producing a vector of realized values $\mathbf{W}_\star^{(i)} = \mathbf{w}$ for the variables in $\mathbf{W}_\star$, the vector is said to be an *i.i.d sample* from $Q$ if $P^{\mathbb{C}}(\mathbf{W}_\star^{(i)} = \mathbf{w} \mid \mathcal{A}^{(i)}) = P^{\mathcal{M}}(\mathbf{W}_\star = \mathbf{w}), \forall \mathbf{w}$, where $P^{\mathbb{C}}$ is the probability measure over the beliefs of the acting agent $\mathbb{C}$, and the l.h.s is the probability of physical actions $\mathcal{A}^{(i)}$ producing the vector $\mathbf{w}$ when performed on some unit $i$. ∎

**Definition 3.4** (Realizability). Given a causal diagram $\mathcal{G}$ and the set of physical actions $\mathbb{A}$, an $\mathcal{L}_3$-distribution $P(\mathbf{W}_\star)$ is *realizable given $\mathbb{A}$ and $\mathcal{G}$* iff there exists a sequence of actions $\mathcal{A}$ from $\mathbb{A}$ by which an agent can draw an i.i.d sample (Def. 3.3) from $P^{\mathcal{M}}(\mathbf{W}_\star)$, for any $\mathcal{M} \in M(\mathcal{G})$, the class of SCMs compatible with $\mathcal{G}$. ∎

We emphasize the *distinction between realizability and identifiability*. Identifiability (Pearl, 2000, Def. 3.2.3) from $\mathcal{G}$ states that a distribution (say, $P(\mathbf{v}; do(x))$) can be uniquely computed from the available data (say, $P(\mathbf{v})$) for any SCM compatible with the assumptions in $\mathcal{G}$. Realizability of a distribution states that it is physically possible for an agent to actually gather data samples according to this distribution.

We next develop an algorithm to decide whether a distribution is realizable. As an intuition pump, suppose that an agent is able to perform CTF-RAND$(V \rightarrow C), \forall V, C \in Ch(V)$, w.r.t an input causal diagram, and wants to obtain samples from $P(Z_x, W_t)$. Consider the diagram $\mathcal{G}_2$ in Fig. 3. By performing CTF-RAND$(T \rightarrow W)$ and CTF-RAND$(X \rightarrow Z)$, the distribution is realizable. However, suppose the input diagram is $\mathcal{G}_1$. A necessary condition to measure $Z_x$ for a unit is for mechanism $f_A$

---

[3]In the philosophy of science literature, similar ideas have been discussed under the topic of the temporal asymmetry of causation (Reichenbach, 1956, §III-IV).

---

**Algorithm 1** CTF-REALIZE

---

1: **Input:** $\mathcal{L}_3$-distribution $Q = P(\mathbf{W}_\star)$; causal diagram $\mathcal{G}$; action set $\mathbb{A}$

2: **Output:** I.i.d sample $\mathbf{W}_\star^{(i)}$ from $Q$; FAIL if $Q$ is not realizable given $\mathcal{G}, \mathbb{A}$

3: Fix a topological ordering $\text{Top}(\mathcal{G})$

4: $\text{SELECT}^{(i)}$ for a new unit $i$

5: **for** $V$ in order $\text{Top}(\mathcal{G})$ **do**

6:    $\text{INT}_V \leftarrow \emptyset$ {Interventions for $V$}

7:    $\text{OUTPUT}_V \leftarrow \emptyset$ {Index in output vector}

8:    **for** each term $W_\mathbf{t}$ in expression $\mathbf{W}_\star$ **do**

9:       **if** $V \in An(W)_{\mathcal{G}_{\overline{\mathbf{T}}}}$ and $V \neq W$ **then**

10:        Call **COMPATIBLE**$(V, W_\mathbf{t})$ Alg. 2

11:       **end if**

12:       **if** $V = W$ **then**

13:        Add $\{W_\mathbf{t}\}$ to $\text{OUTPUT}_V$

14:       **end if**

15:    **end for**

16:    **for** each $\{\text{action} : \text{tag}\} \in \text{INT}_V$ **do**

17:       Perform the randomization on unit $i$

18:       If the random-generated value $\neq$ tag, discard the unit and return to Line 4

19:    **end for**

20:    **for** each $W_t \in \text{OUTPUT}_V$ **do**

21:       **if** $\{\text{RAND}(V) : .\} \in \text{INT}_V$ **then**

22:        Return FAIL

23:       **else**

24:        Perform $\text{READ}(V)^{(i)} = v'$

25:        Assign $v'$ to each index $W_t^{(i)}$ in output vector $\mathbf{W}_\star^{(i)} = \mathbf{w}$

26:       **end if**

27:    **end for**

28: **end for**

29: Return i.i.d sample $\mathbf{W}_\star^{(i)} = \mathbf{w}$

---

to receive the natural value of $T$, illustrated in green. While a necessary condition to simultaneously measure $W_t$ is for $f_W$ to receive $A_t$, which in turn requires $f_A$ to receive a fixed $t$, shown in red. This conflict in necessary conditions renders the query non-realizable.[4]

This "edge-coloring" intuition is formalized in Algo. 1. The algorithm **CTF-REALIZE** takes as input an $\mathcal{L}_3$-distribution $P(\mathbf{W}_\star)$, a graph $\mathcal{G}$, and a set of physical actions $\mathbb{A}$ the agent is able to perform in the environment (viz., the RAND and CTF-RAND actions which are possible in the environment). It returns an i.i.d sample if the distribution is realizable, and FAIL otherwise.

The algorithm works as follows (a more detailed walkthrough is presented in App. C.2 of the full technical report): going over each node $V$ in topological order, the inner loops gather the necessary and sufficient conditions

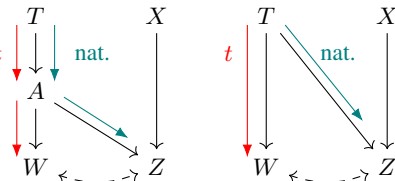

Figure 3: Testing realizability of $P(Z_x, W_t)$ for $\mathcal{G}_1$ (left) and $\mathcal{G}_2$ (right). $\mathcal{G}_1$ yields conflicting requirements.

needed w.r.t $V$ for realizing each $W_\mathbf{t}$ in the input query $\mathbf{W}_\star$. If there is a conflict in the necessary conditions for evaluating two terms (as we saw for $P(Z_x, W_t)$ in Fig. 3, $\mathcal{G}_1$), the query is non-realizable. The algorithm is fully general and does not make assumptions about the ability to perform any particular interventions. If the agent cannot perform any counterfactual randomization, the algorithm returns FAIL for non-$\mathcal{L}_2$ queries. If the agent cannot perform any interventions at all, the algorithm returns FAIL for non-$\mathcal{L}_1$ queries (we assume the ability to READ all variables). Details about the time and space complexity of Algo. 1 are provided in App. C.3 of the technical report.

**Theorem 3.5** (Correctness and Completeness). *An $\mathcal{L}_3$-distribution $Q = P(\mathbf{W}_\star)$ is realizable given action set $\mathbb{A}$ and causal diagram $\mathcal{G}$ iff the algorithm **CTF-REALIZE**$(Q, \mathcal{G}, \mathbb{A})$ returns a sample.* ∎

A further question we may ask is which $\mathcal{L}_3$-distributions are realizable if we assume maximum experimental capabilities, notably, the ability to perform separate ctf-randomization for each child of each variable. Given a causal diagram $\mathcal{G}$, we define the *maximal feasible action set* $\mathbb{A}^\dagger(\mathcal{G})$ as the set containing all of the following actions: $\text{SELECT}^{(i)}$, $\text{READ}(V)^{(i)}$, $\forall V$, and $\text{CTF-RAND}(X \to C)^{(i)}$, $\forall X$ and $C \in Ch(X)$. $\mathbb{A}^\dagger(\mathcal{G})$ thus gives the agent the most granular interventional capabilities.

**Definition 3.6** (Ancestors of a counterfactual (Correa et al., 2021)). Given a causal diagram $\mathcal{G}$ and a potential response $Y_\mathbf{x}$, the set of (counterfactual) ancestors of $Y_\mathbf{x}$, denoted $An(Y_\mathbf{x})$, consists of each

---

[4]To be clear, the input to the algorithm is a graph and an accurate set of actions the agent can perform in the environment. If the graph is per $\mathcal{G}_1$ in Fig. 3, then CTF-RAND$(T \to Z)$ is not possible in this environment. Marginalizing out $A$ and providing graph $\mathcal{G}_2$ as input does not help.

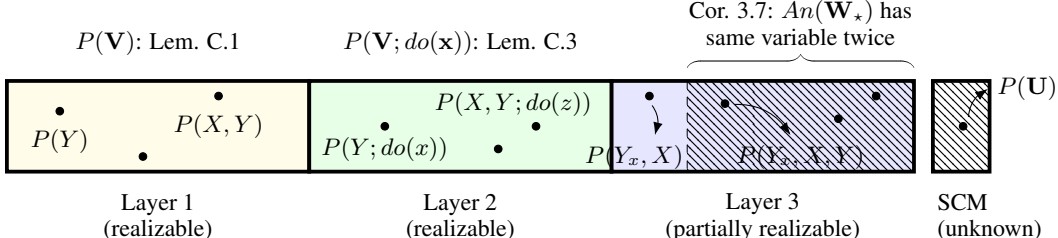

Figure 4: Pearl Causal Hierarchy (PCH) induced by an unknown SCM $\mathcal{M}$. An $\mathcal{L}_3$-distribution is realizable given a graph $\mathcal{G}$ and the maximal feasible action set $\mathbb{A}^\dagger(\mathcal{G})$ iff the ancestor set $An(\mathbf{W}_\star)$ does not contain the same variable under different regimes.

$W_{\mathbf{z}}$ s.t. $W \in An(Y)_{\mathcal{G}_{\underline{\mathbf{X}}}}$, and $\mathbf{z} = \mathbf{x} \cap An(W)_{\mathcal{G}_{\overline{\mathbf{X}}}}$. For a set $\mathbf{W}_\star$, $An(\mathbf{W}_\star)$ is defined to be the union of the ancestors of each potential response in the set. ∎

**Corollary 3.7.** *An $\mathcal{L}_3$-distribution $Q = P(\mathbf{W}_\star)$ is realizable given causal diagram $\mathcal{G}$ and action set $\mathbb{A}^\dagger(\mathcal{G})$ iff the ancestor set $An(\mathbf{W}_\star)$ does not contain a pair of potential responses $W_{\mathbf{t}}, W_{\mathbf{s}}$ of the same variable $W$ under different regimes.* ∎

For instance, if $\mathbf{W}_\star = \{Z_x, W_t\}$ w.r.t graph $\mathcal{G}_1$ in Fig. 3, then $An(\mathbf{W}_\star) = \{Z_x, A, T, W_t, A_t\}$, which contains both $A, A_t$. Thus, $P(\mathbf{W}_\star)$ is not realizable even with maximal experimentation capabilities. In App. C.4 of the technical report, we provide further examples of using the **CTF-REALIZE** algorithm, and the graphical criterion, to demonstrate the realizability of the ETT distribution $P(Y_x, X)$, the non-realizability of the probability of sufficiency distribution $P(Y_x, X, Y)$.

We believe this is an important contribution to causal inference. Cor. 3.7 provides a graphical criterion to delineate how far up the PCH an agent can go via experimental methods, in principle. Often, counterfactuals have been criticized as being hypothetical, untestable, or unscientific assumptions. Our analysis counters this claim, as summarized in Fig. 4.

**Corollary 3.8** (Fundamental problem of causal inference (FPCI) (Holland, 1986))**.** *The distribution $Q = P(Y_x, Y_{x'})$ is not realizable given maximal feasible action set $\mathbb{A}^\dagger(\mathcal{G})$, for any causal diagram $\mathcal{G}$, and any variables $X, Y \in Desc(X)$.* ∎

The FPCI is an influential notion in the literature, and is often taken as a primitive, or in an axiomatic fashion. We show that it is rather a specific consequence of the more general FCE assumption 3.1, and follows from Thm. 3.5 and Cor. 3.7. By itself, the FPCI does not translate to an operational criterion for determining which $\mathcal{L}_3$-distributions are realizable (Def. 3.4). For instance, it does not clarify that a distribution with potential responses under conflicting regimes like $P(Y_x, Z_{x'})$ may indeed be realizable via counterfactual randomization, as we show in Example 2. It also does not tell us that $P(Z_x, W_t)$ may be realizable given causal diagram $\mathcal{G}_2$ in Fig. 3, but not realizable given $\mathcal{G}_1$.

# 4 APPLICATIONS: COUNTERFACTUAL DECISION-MAKING AND FAIRNESS

Next, we highlight the practical relevance of our results with some concrete use-cases. We already discussed in Example 1 how realizability can be used to design experiments for performing **mediation analysis** of direct and indirect effects, an important task in several fields. We now discuss applications in **causal fairness analysis** and **causal reinforcement learning (RL)**. Our goal is to underscore that the standard/baseline approaches in these areas, even among approaches that incorporate counterfactual reasoning, typically use observational ($\mathcal{L}_1$) or interventional ($\mathcal{L}_2$) data only, whereas a counterfactual ($\mathcal{L}_3$) data-collection approach can lead to demonstrably better results. We include in App. F of the full technical report the specification of SCMs used and algorithms implemented.

## 4.1 CAUSAL FAIRNESS - USING COUNTERFACTUAL DATA FOR FAIRER DECISIONS

Causal fairness analysis is a burgeoning field and a full survey is beyond the scope of this paper (see, e.g., Plecko & Bareinboim (2024) for a review of related works). We limit our discussion to an example where counterfactual realizability is directly relevant.

A common concern is that models trained to make automated decisions often reveal problematic biases (Angwin et al., 2016; Kodiyan, 2019, e.g.). The causal approach to address this is typically to constrain a classifier to obey some causally-sensitive *fairness measure*, $\mu$ (Plecko & Bareinboim, 2024, Def. 3.3). Some measures in the literature involve $\mathcal{L}_3$-quantities, and thus face the familiar issue of nonidentifiability (Kusner et al., 2017; Imai & Jiang, 2023). Other approaches acknowledge this limitation and try to construct interventional fairness measures that solely use $\mathcal{L}_2$-quantities (Salimi et al., 2019). We present next an example where relying only on $\mathcal{L}_2$-data can misleadingly approve a classifier as fair, but where a realizable $\mathcal{L}_3$ fairness measure actually ensures fairness. This scenario is inspired by a classic experiment in labor economics (Bertrand & Mullainathan, 2003).

**Example 2 (Causal fairness).**   A college is developing an automated system to screen candidates in the first round of college applications, receiving as input a standardized CV per candidate. The system contains two models: model 1 outputs $Y$ and model 2 outputs $Z$, which are binary decisions of whether the applicant cleared the first review stage for admission and for financial scholarship, respectively. The two models are respectively trained using data from previous years where an admissions team and a separate scholarship team reviewed applications manually. The college wants to ensure fairness w.r.t $X$, a candidate's race (a binary variable, for simplicity). In particular, they want to ensure equitable financial access to education for all qualified candidates: a candidate of race $X = 1$ who cleared the admissions screening ($Y = 1$) but was rejected for financial aid ($Z = 0$) should still receive $Z = 0$ had they been of race $X = 0$. The causal diagram is in Fig. 5(a), where the models' decisions $Y, Z$ might reflect the unconscious race bias of the two committees in previous years (including possibly shared biases, represented by the latent confounder).

The $\mathcal{L}_3$ fairness measure they ought to minimize is thus

$$\mu_{ctf} = |P(Y_{x_1} = 1, Z_{x_1} = 0) - P(Y_{x_1} = 1, Z_{x_0} = 0)| \quad (6)$$

But the second term $P(y_x, z'_{x'})$ is nonidentifiable from the causal diagram in 5(a). So the college instead uses the following $\mathcal{L}_2$ measures, as an approximation for the fairness condition:

$$\mu_{int1} = |P(Y = 1; do(x_1)).P(Z = 0; do(x_1)) \quad (7)$$
$$- P(Y = 1; do(x_1)).P(Z = 0; do(x_0))|$$
$$\mu_{int2} = |P(Y = 1, Z = 0; do(x_1)) \quad (8)$$
$$- P(Y = 1, Z = 0; do(x_0))|$$

They train the models, adding $\mu_{int1} + \mu_{int2}$ as a penalty in the objective. $\mu_{int1}, \mu_{int2}$ are estimated using a holdout set of fake CVs, with the intervention $do(x)$ being enacted by randomly choosing an applicant name from an equivalence class which stereotypically indicates one unique race group $X = x$, e.g. names like Lakisha and Jamal for Blacks, or last names like Nguyen or Xi for Asians (cf. Bertrand & Mullainathan (2003)). Since the holdout set's CV body is independent of $X$, any effect of $X$ on $Y$ and $Z$ is solely via the perception of race from the candidate name. We show in 5(c) simulations of such an optimization. In blue is the distribution of the true score $\mu_{ctf}$, when the models are trained using $\mu_{int1}, \mu_{int2}$. Out of 1000 simulations of classifiers, we see $\mu_{ctf} > 5\%$ for nearly half the $\mathcal{L}_2$ simulations, indicating statistically significant discrimination roughly 50% the time.

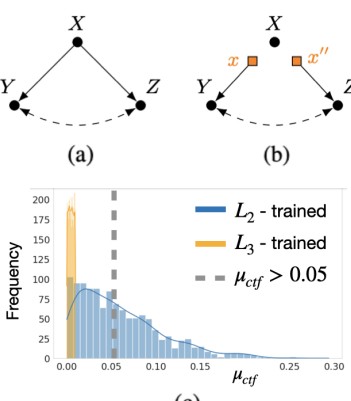

Figure 5: (a) Causal diagram for Example 2; (b) $P(Y_x, Z_{x'})$ is realizable using the interventions CTF-RAND($X \rightarrow Y$) and CTF-RAND($X \rightarrow Z$); (c) Histogram of 1000 classifiers trained on $\mathcal{L}_2$ (blue) and $\mathcal{L}_3$ (orange) fairness measures. $\mathcal{L}_2$ classifiers show statisically significant discrimination ($\mu_{ctf} > 0.05$).

However, the distribution $P(Y_x, Z_{x'})$ is indeed realizable (Def. 3.4) via the interventions CTF-RAND($X \rightarrow Y$), CTF-RAND($X \rightarrow Z$). The data science team notices that they can *separately and simultaneously* randomize the candidate name as an input to the respective models, and enact these interventions, as shown in 5(b). Thus, they are able to directly use the counterfactual measure $\mu_{ctf}$ as a fairness constraint in training. Results from 1000 simulations show that the classifiers trained directly using $\mu_{ctf}$ (shown in orange) nearly always meet the fairness requirement.

Details of the implementation are in App. F.2 here. Note: as in the original experiment, this example requires the structural assumption of race being revealed at the screening stage only by candidate name, which may be more defensible in highly standardized and controlled application processes. ∎

## 4.2 CAUSAL RL - COUNTERFACTUAL POLICIES FOR OPTIMAL DECISION-MAKING

Consider a multi-arm bandit problem in which $X$ represents the choice of bandit arm and $Y$ the outcome. The default online learning approach is for the agent to adopt an algorithm like EXP3, UCB or Thompson Sampling to converge to some arm $x^\star := \arg\max_x \mathbb{E}[Y; do(x)]$ (Lattimore & Szepesvári, 2020; Sutton & Barto, 1998). Even in methods that explicitly incorporate causal knowledge, the common approach is to use a combination of offline ($\mathcal{L}_1$) and online ($\mathcal{L}_2$) data to converge more efficiently to the $\mathcal{L}_2$ optimization target $\arg\max_x \mathbb{E}[Y; do(x)]$ (Zhang & Bareinboim, 2017, e.g.). It was already shown in (Bareinboim et al., 2015; Forney et al., 2017) that it is possible to perform better by deploying a counterfactual strategy based on sampling each unit's natural choice $X = x'$ and randomizing actual choice in the *same* round, thus seeking to converge to $\arg\max_x \mathbb{E}[Y_x \mid x'], \forall x'$, as we discussed in Sec. 2. We call this the ETT baseline strategy, as it relies on drawing samples from the $\mathcal{L}_3$ ETT distribution, $P(Y_x, X)$, mentioned in Sec. 1.

We improve on this baseline by showing how an agent can leverage the realizability (Def. 3.4) of more nuanced counterfactuals like $P(Y_x, X, D_{x''})$ to construct superior counterfactual strategies. The following scenario involves an agent faced with adversarial latent confounding.

**Example 3 (Counterfactual bandit policies).** Consider a user of a social media platform which uses surveillance and predictions to increase user engagement through addictive notifications and recommendations (Zuboff, 2018). The user chooses every evening whether to use the platform via desktop ($X = 0$) or mobile ($X = 1$). $Y$ is a binary indicator of whether she stays within her self-determined social media usage limit per day. She also notices that she receives ads when she logs in each evening as $D$ (0: streaming service, 1: food delivery ads). The usage type $X$ affects $D, Y$, as shown in Fig. 6(a).

On average, the user experiences $\mathbb{E}[Y] = 0.65$ from the observational ($\mathcal{L}_1$) policy of following her natural inclination each day. She suspects that the company could be tracking and exploiting her latent preferences, so she decides to randomize her daily choice and pick the best

Table 1: Performance of different strategies in Example 3.

| Strategy | $\mathbb{E}[Y]$ |
|---|---|
| Behavioral policy ($\mathcal{L}_1$) | 0.65 |
| Naive randomization ($\mathcal{L}_2$) | 0.7 |
| ETT baseline strategy ($\mathcal{L}_3$) | 0.75 |
| Optimal $\mathcal{L}_3$ strategy (this work) | 0.80 |

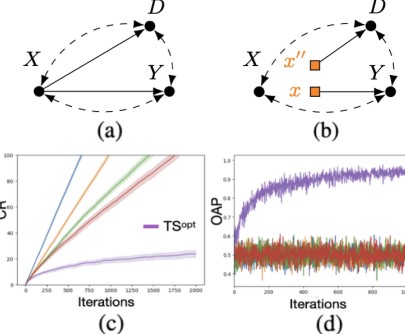

Figure 6: (a) Causal diagram for Example 3; (b) $P(Y_x, X, D_{x''})$ is realizable using the interventions CTF-RAND($X \to Y$) and CTF-RAND($X \to D$); (c) Cumulative Regret (CR) for $\mathcal{L}_1$ strategy (blue) and Thompson Sampling algorithms implementing naive $\mathcal{L}_2$ (yellow, green), ETT baseline (red), and optimal $\mathcal{L}_3$ strategy (purple); (d) Optimal Arm Probability (OAP) for all algorithms.

"arm". Sure enough, this naive $\mathcal{L}_2$ strategy breaks the adversarial confounding, and incurs a better avg. performance of $\mathbb{E}[Y; do(x)] = 0.7, \forall x$. She then decides to test the ETT-based strategy ($\mathcal{L}_3$) described earlier, by recording what she naturally feels like doing each day ($X = x'$), and subsequently randomizing her actual choice on the same day to optimize $\mathbb{E}[Y_x \mid x']$, getting an avg. performance of 0.75. However, at this point, she notices that she can do even better. The $\mathcal{L}_3$-distribution $P(Y_x, X, D_{x''})$ is realizable (Def. 3.4), since she can perform *another* counterfactual randomization, by sampling her natural choice ($X = x'$), randomly logging in to just see what ads she gets ($D_{x''} = d$), and again randomizing how she actually uses the platform that day to get $Y_x$. This strategy seeks an optimal $x^\star = \arg\max_x \mathbb{E}[Y_x \mid x', d_{x''}]$, which performs best as shown in Table 1. Details of the SCM, latent confounders, and the optimal $\mathcal{L}_3$-strategy are in App. F.3 here.

Simulations in the online setting corroborate this finding. Fig. 6(c,d) shows the cumulative regret (CR) and optimal arm probability (OAP) over 2000 iterations averaged over 200 epochs (CI=95%). We adapt Thompson Sampling to implement the strategies in Table 1. Details of implementation are in App. F.3.1 here. The optimal $\mathcal{L}_3$ strategy (purple) performs best, improving on the performance of the baseline ETT-based strategy (red). Naive randomizations, the standard $\mathcal{L}_2$ bandit strategy, are shown in yellow and green. All other algorithms fail to improve in OAP after 2000 iterations. ∎

We make two remarks. First, the optimal counterfactual strategy is not simply a contextual Thompson Sampling, where $X, D_{x''}$ are used "merely" as extra context variables per round; indeed, treating this merely as a contextual bandit problem is one of the naive $\mathcal{L}_2$-strategies that we test (green plot in Fig. 6(c-d)), which ignores the counterfactual relationship between these variables and incurs dramatically higher regret, as we discuss in App. F.3.1 here.

Second, an interesting follow-up is whether we can guarantee that our strategy based on maximizing $\mathbb{E}[Y_x \mid x', d_{x''}]$ is optimal in this problem. Perhaps there are more refined $\mathcal{L}_3$-distributions like $P(Y_x, X, D_x, D_{x''})$ etc. that could yield better algorithms? It turns that this is indeed optimal, since most other $\mathcal{L}_3$-distributions are not realizable (Def. 3.4). We prove this claim for all bandit problems that fit a generic template, in Thm. 4.2 of the technical report. Thereby, we avoid having to conduct an intractable search over the space of all possible $\mathcal{L}_3$-strategies, trying to assess their realizability.

## 5 DISCUSSION

Finally, we discuss some important implications, future directions, and limitations of our work.

**Identification and bounding.** Much work has been done in the area of $\mathcal{L}_3$ identification and estimation (Shpitser, 2008; Correa et al., 2021; Geneletti & Dawid, 2011). A natural extension to our work is to investigate the relationship between realizability and identification: which *additional* $\mathcal{L}_3$-quantities now become identifiable if the environment permits even some counterfactual randomization? This warrants an update to existing identification algorithms to allow (some) $\mathcal{L}_3$-data as input. Another fascinating research question involves "partial identification", where an input query is tightly bounded within a range that can be computed from available data (Zhang et al., 2022): how would the new $\mathcal{L}_3$-data further tighten the bounds for nonidentifiable $\mathcal{L}_3$-quantities?

**Experiment design.** One of the goals of this paper is to instigate new experiment design ideas that leverage ctf-randomization (Def. 2.3) and go beyond the standard RCT methodology, as in Examples 1-2. For instance, the increasingly automated HR pipeline in companies suggests opportunities for targeted interventions to randomize demographic details in virtual interviews, in standardized aptitude tests, or in performance-evaluation systems for remote workers, to track fairness metrics.

**Causal reinforcement learning (CRL).** While counterfactual strategies have been studied in CRL, the literature currently focuses on ETT-related strategies based on optimizing $\mathbb{E}[Y_x \mid x']$ (Bareinboim et al., 2015; Forney et al., 2017; Zhang & Bareinboim, 2022)(Richardson & Robins, 2013, §5.1). We presented an important extension by formalizing ctf-randomization (Def. 2.3) via counterfactual mediators (Def. 2.2), subsuming the previous approach. An ETT-based approach only allows one randomization of a variable $X$, affecting all downstream mechanisms. Our approach recognizes the possibility of isolating specific causal pathways and randomizing $X$ multiple times per unit, demonstrably surpassing the ETT baseline in Example 3. We proved in Thm. 4.2 of the technical report an optimality guarantee for our proposed strategy in bandit problems. Generalizing this to sequential decision-making settings with arbitrary graphs is an important, non-trivial extension.

**Limitations.** The first obvious limitation of our framework is that it requires causal knowledge in the form of a graph (or equivalent). This is a standard assumption, needed to make progress in several areas of causal machine learning. Subsequent work could accommodate partial knowledge or model misspecification. Second, it may not always be feasible to perform counterfactual randomization (Def. 2.3) in a given setting. This is why Algo. 1 and Thm. 3.5 are general and do not assume this capability a priori. But where it is possible, even in principle, our work pinpoints opportunities for novel experiment design, as discussed above.

## 6 CONCLUSION

In this paper, we tackle the open question of which counterfactual distributions are directly accessible by experimental methods - what we define as the *realizability* of a distribution. Countering prevalent belief, we provide a complete algorithm and a graphical criterion for when a counterfactual can indeed be physically sampled from (Fig. 4). We demonstrate the practical relevance of this new framework with examples from causal fairness and causal RL, highlighting that ignoring this possibility could lead to poor outcomes. We believe that switching from an interventional to a counterfactual mindset could help researchers spot opportunities for *counterfactual randomization* that permit exciting new types of experiments, and improved, more personalized decisions.

ACKNOWLEDGEMENTS

This research is supported in part by the NSF, ONR, AFOSR, DoE, Amazon, JP Morgan, and The Alfred P. Sloan Foundation. We thank Juan D. Correa and the anonymous reviewers for their thoughtful comments.

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

## A  GRAPHICAL TERMINOLOGY

Structural Causal Models (SCM) and causal diagrams are described in the preliminaries in Sec. 1. See (Bareinboim et al., 2022) for full treatment. We use the following graphical kinship nomenclature w.r.t causal diagram $\mathcal{G}$:

- Parent(s) of $V$, denoted $\mathbf{Pa}_V$: the set of variables $\{V'\}$ s.t. there is a direct edge $V' \to V$ in $\mathcal{G}$. $\mathbf{Pa}_V$ does not include $V$.

- Children of $V$, denoted $Ch(V)$: the set of variables $\{V'\}$ s.t. there is a direct edge $V \to V'$ in $\mathcal{G}$. $Ch(V)$ does not include $V$.

- Ancestors of $V$, denoted $An(V)$: the set of variables $\{V'\}$ s.t. there is a path (possibly length 0) from $V'$ to $V$ consisting only of edges pointing toward $V$, $V' \to ... \to V$. $An(V)$ is defined to include $V$.

- Descendants of $V$, denoted $Desc(V)$: the set of variables $\{V'\}$ s.t. there is a path (possibly length 0) from $V$ to $V'$ consisting only of edges pointing toward $V'$, $V \to ... \to V'$. $Desc(V)$ is defined to include $V$.

- Non-descendants of $V$, denoted $NDesc(V)$: the set $\mathbf{V} \setminus Desc(V)$. $NDesc(V)$ does not include $V$.

Given a graph $\mathcal{G}$, $\mathcal{G}_{\overline{\mathbf{X}}\underline{\mathbf{W}}}$ is the result of removing edges coming into variables in $\mathbf{X}$, and edges coming out of $\mathbf{W}$.

## B  SUB-ROUTINE OF **CTF-REALIZE** ALGORITHM (ALGO. 1)

---

**Algorithm 2** COMPATIBLE (sub-routine)

---

1: **Input:** $V \in \mathbf{V}$ of $\mathcal{G}$; $W_{\mathbf{t}} \in \mathbf{W}_\star$ of $Q$
2: **for** each $C \in Ch(V)$ **do**
3:   **if** $C \in An(W)$ **then**
4:     **if** $V \in \mathbf{T}$ **then**
5:       Let $v :=$ value of $V$ in subscript $\mathbf{t}$
6:       Find smallest $\mathbf{C} \ni C$ s.t. CTF-RAND$(V \to \mathbf{C}) \in \mathbb{A}$
7:       **if** $\{$CTF-RAND$(V \to \mathbf{C}) : .\} \in$ INT$_V$ and its label is not "$v$" **then**
8:         Return FAIL
9:       **else**
10:         Add $\{$CTF-RAND$(V \to \mathbf{C}) : v\}$ to INT$_V$, with the label "$v$"
11:       **end if**
12:       **if** no such $\mathbf{C} \ni C$ s.t. CTF-RAND$(V \to \mathbf{C}) \in \mathbb{A}$ **then**
13:         **if** $\{$RAND$(V) : .\} \in$ INT$_V$ and its label is not "$v$" **then**
14:           Return FAIL
15:         **else if** RAND$(V) \notin \mathbb{A}$ **then**
16:           Return FAIL
17:         **else**
18:           Add $\{$RAND$(V) : v\}$ to INT$_V$, with the label "$v$"
19:         **end if**
20:       **end if**
21:     **end if**
22:     **if** $V \notin \mathbf{T}$ **then**
23:       **for** each $\mathbf{C} \ni C$ s.t. CTF-RAND$(V \to \mathbf{C}) \in \mathbb{A}$ **do**
24:         **if** $\{$CTF-RAND$(V \to \mathbf{C}) : .\} \in$ INT$_V$ and its label is not "Natural" **then**
25:           Return FAIL
26:         **else**
27:           Add $\{$CTF-RAND$(V \to \mathbf{C}) :$ Natural$\}$ to INT$_V$, with the label "Natural"
28:         **end if**
29:       **end for**
30:       **if** $\{$RAND$(V) : .\} \in$ INT$_V$ and its label is not "Natural" **then**
31:         Return FAIL
32:       **else if** RAND$(V) \in \mathbb{A}$ **then**
33:         Add $\{$RAND$(V) :$ Natural$\}$ to INT$_V$, with the label "Natural"
34:       **end if**
35:     **end if**
36:   **end if**
37: **end for**

---

For a detailed walkthrough of the algorithm and sub-routine, refer to Appendix C.2 of the full technical report here.

The full technical report also contains detailed examples of applying the algorithm to different graphs and queries, in its Appendix C.4 (Raghavan & Bareinboim, 2025).

## C ASSUMPTIONS AND REALIZABILITY PROOFS

For a summary of all the structural assumptions we make in this paper, and proofs of the results, refer to Appendix D of the full technical report here.

### C.1 REALIZABILITY OF $\mathcal{L}_1$- AND $\mathcal{L}_2$-DISTRIBUTIONS

It is widely known and acknowledged that it is possible to draw samples from $\mathcal{L}_1$- and $\mathcal{L}_2$-distributions: the former by simply observing a system's natural behaviour, and the latter by intervening in the system through interventions like Fisherian randomization.

Still, we find it educational to derive these proofs from first principles. This sub-section is not strictly needed to follow the main contributions in Secs. 2 and 3.

We define the probability measure $P^{\mathbb{C}}(.)$ from the perspective of an exogenous agent (i.e., an agent external to the system) $\mathbb{C}$'s beliefs about the environment, distinguished by superscript from $P^{\mathcal{M}}(.)$, the true unknown distribution.

Since unit selection is randomized, $\text{SELECT}^{(i)}$ yields an unbiased sample of a unit with latent features distributed according to the target population frequency $P(\mathbf{u})$. I.e., $P^{\mathbb{C}}(\mathbf{U}^{(i)} = \mathbf{u} \mid \text{SELECT}^{(i)}) = P^{\mathcal{M}}(\mathbf{u})$. We also assume that target population size is large enough that $\text{SELECT}^{(i)}$ does not significantly change the distribution of the remaining population.

Further, we assume that the actions $\text{READ}(V)^{(i)}$ and $\text{RAND}(V)^{(i)}$ do not disrupt any other mechanism $f_{V'}$ for unit $i$.

**Lemma C.1** (Observational sample). *An agent $\mathbb{C}$ can draw an i.i.d sample distributed according to the $\mathcal{L}_1$ query $P(\mathbf{V})$ associated with an SCM $\mathcal{M}$, by the following actions:*

*i.* $\text{SELECT}^{(i)}$

*ii.* $\text{READ}(\mathbf{V})^{(i)} = \mathbf{v} \sim P(\mathbf{V})$

*Given $N$ i.i.d samples, the consistent unbiased estimate of $P(\mathbf{v})$ is*

$$\hat{P}(\mathbf{v}) := \frac{1}{N} \sum_i \prod_{v \in \mathbf{v}} \mathbb{1}[\text{READ}(V)^{(i)} = v] \tag{9}$$

*Proof.* This follows directly from the definitions of the actions. $\text{SELECT}^{(i)}$ chooses a unit at random from the population. By Remark D.3, $P^{\mathbb{C}}(\mathbf{U}^{(i)} = \mathbf{u} \mid \text{SELECT}^{(i)}) = P^{\mathcal{M}}(\mathbf{u})$. For randomly selected unit $i$,

$$P^{\mathbb{C}}(\text{READ}(\mathbf{V})^{(i)} = \mathbf{v} \mid \text{SELECT}^{(i)}) \tag{10}$$

$$= \sum_{\mathbf{u}} P^{\mathbb{C}}(\mathbf{U}^{(i)} = \mathbf{u} \mid \text{SELECT}^{(i)}). \tag{11}$$

$$P^{\mathbb{C}}(\text{READ}(\mathbf{V})^{(i)} = \mathbf{v} \mid \mathbf{U}^{(i)} = \mathbf{u}, \text{SELECT}^{(i)}) \qquad \text{Chain rule}$$

$$= \sum_{\mathbf{u}} P^{\mathbb{C}}(\mathbf{U}^{(i)} = \mathbf{u} \mid \text{SELECT}^{(i)}).\mathbb{1}^{\mathcal{M}}[\mathbf{V}(\mathbf{u}) = \mathbf{v}] \qquad \text{Def. 2.1(ii)} \tag{12}$$

$$= \sum_{\mathbf{u}} P^{\mathcal{M}}(\mathbf{u}).\mathbb{1}^{\mathcal{M}}[\mathbf{V}(\mathbf{u}) = \mathbf{v}] \qquad \text{Rem. D.3} \tag{13}$$

$$= P^{\mathcal{M}}(\mathbf{v}) \qquad \text{Definition} \tag{14}$$

I.e., this record is an i.i.d. sample from $P^{\mathcal{M}}(\mathbf{V})$. Now consider the estimator below.

$$\hat{P}(\mathbf{v}) := \frac{1}{N} \sum_n \prod_{v \in \mathbf{v}} \mathbb{1}^{\mathbb{C}}[\text{READ}(V)^{(i)} = v] \tag{15}$$

$$= \frac{1}{N} \sum_n \sum_{\mathbf{u}} \prod_{v \in \mathbf{v}} \mathbb{1}^{\mathcal{M}}[\mathbf{U}^{(i)} = \mathbf{u}].\mathbb{1}^{\mathcal{M}}[V(\mathbf{u}) = v] \tag{16}$$

Un-biasedness is established by taking expectation on either side, w.r.t the agent $\mathbb{C}$'s actions (choice of units to observe):

$$\mathbb{E}_{\mathbb{C}}[\hat{P}(\mathbf{v})] = \mathbb{E}_{\mathbb{C}}\left[ \frac{1}{N} \sum_n \sum_{\mathbf{u}} \prod_{v \in \mathbf{v}} \mathbb{1}^{\mathcal{M}}[\mathbf{U}^{(i)} = \mathbf{u}].\mathbb{1}^{\mathcal{M}}[V(\mathbf{u}) = v] \right] \tag{17}$$

$$= \sum_{\mathbf{u}} \frac{1}{N} \mathbb{E}_{\mathbb{C}}\left[ \sum_n \mathbb{1}^{\mathcal{M}}[\mathbf{U}^{(i)} = \mathbf{u}] \prod_{v \in \mathbf{v}} .\mathbb{1}^{\mathcal{M}}[V(\mathbf{u}) = v] \right] \quad \text{Linearity of expectation} \tag{18}$$

$$= \sum_{\mathbf{u}} \frac{1}{N} \mathbb{E}_{\mathbb{C}}\left[ \sum_n \mathbb{1}^{\mathcal{M}}[\mathbf{U}^{(i)} = \mathbf{u}] \right] \prod_{v \in \mathbf{v}} \mathbb{1}^{\mathcal{M}}[V(\mathbf{u}) = v] \quad V(\mathbf{u}) \text{ constant wrt } \mathbb{C} \tag{19}$$

$$= \sum_{\mathbf{u}} \frac{1}{N} \left[ N.P^{\mathcal{M}}(\mathbf{u}) \right] \prod_{v \in \mathbf{v}} \mathbb{I}^{\mathcal{M}}[V(\mathbf{u}) = v] \quad \text{Def. 2.1(i), Rem. D.3} \tag{20}$$

$$= P^{\mathcal{M}}(\mathbf{v}) \quad \text{Definition} \tag{21}$$

Consistency is established by the fact that as $\mathcal{N}$ (target population size) $\to \infty$, and $N$ (sample size) $\to \infty$,

$$\frac{1}{N} \sum_n \mathbb{I}^{\mathcal{M}}[\mathbf{U}^{(i)} = \mathbf{u}] \to P^{\mathcal{M}}(\mathbf{u}) \tag{22}$$

$\blacksquare$

**Lemma C.2.** *The $\mathcal{L}_2$ distribution of an atomic intervention is equivalent to the $\mathcal{L}_2$ distribution of the corresponding conditional stochastic intervention.*

$$P^{\mathcal{M}}(\mathbf{v}; do(\mathbf{x})) = P^{\mathcal{M}}(\mathbf{v}|\mathbf{x}; \sigma_{\mathbf{X}}) \tag{23}$$

$$= \sum_{\mathbf{u}} \underbrace{\mathbb{1}[\mathbf{V}_{\sigma_{\mathbf{X}}}(\mathbf{u}) = \mathbf{v} \mid X_{\sigma_{\mathbf{X}}} = \mathbf{x}]}_{\text{①}} . \underbrace{P(\mathbf{u})}_{\text{②}} \tag{24}$$

*Proof.* The step from the r.h.s of Eq. 23 to Eq. 24 is derived as follows: in the submodel $\mathcal{M}_{\sigma_{\mathbf{X}}}$, if we are given that $\mathbf{X}$ has been randomly assigned $\mathbf{x}$, then the remaining variables are deterministically generated as a function of $\mathbf{u}$ and $\mathbf{x}$ via their respective equations. The probability mass is collected over all the $\mathbf{u}$ which produce the output $\mathbf{v}$ over all these equations.

$$P^{\mathcal{M}}(\mathbf{v}|\mathbf{x}; \sigma_{\mathbf{X}}) = \sum_{\mathbf{u}} \mathbb{I}[\mathbf{V}_{\sigma_{\mathbf{X}}}(\mathbf{u}) = \mathbf{v} \mid X_{\sigma_{\mathbf{X}}} = \mathbf{x}].P^{\mathcal{M}}(\mathbf{u}) \tag{25}$$

Notice: if $\mathbf{v}$ is incompatible with $\mathbf{x}$, the indicator in the r.h.s evaluates to 0. Next, we prove. Eq. 23.

In $\mathcal{M}_{\sigma_{\mathbf{X}}}$, as defined, $\mathbf{X}$ is assigned according to an independent random vector. Notate this vector as $\mathbf{X}_{\sigma_{\mathbf{x}}}$ and let the distribution of this vector be $P_{\sigma_{\mathbf{x}}}(\mathbf{X})$, defined by the assignment frequency over the target population.

$\mathcal{M}_{\sigma_{\mathbf{X}}}$ is defined such that the target population is split into groups, each assigned $(X_{\sigma_{\mathbf{x}}} = \mathbf{x})$ for some $\mathbf{x}$. Note, the assignment vector $\mathbf{X}_{\sigma_{\mathbf{x}}}$ is independent of the latent features $\mathbf{U}$ across the target population iff each finite group assigned $(X_{\sigma_{\mathbf{x}}} = \mathbf{x})$ has the same distribution of latent features $P(\mathbf{U})$ as in the overall target population.

The above discussion handles the finite size of the target population. Starting with the r.h.s of Eq. 23,

$$P^{\mathcal{M}}(\mathbf{v}|\mathbf{x}; \sigma_{\mathbf{X}}) = \frac{P(\mathbf{v}, \mathbf{x}; \sigma_{\mathbf{X}})}{P(\mathbf{x}; \sigma_{\mathbf{X}})} = \begin{cases} P(\mathbf{v}; \sigma_{\mathbf{X}})/P(\mathbf{x}; \sigma_{\mathbf{X}}) & \text{if } \mathbf{v} \text{ compatible with } \mathbf{x} \\ 0 & \text{otherwise} \end{cases} \tag{26}$$

Evaluating for when $\mathbf{v}$ is compatible with $\mathbf{x}$:

$$\frac{P(\mathbf{v}; \sigma_{\mathbf{X}})}{P(\mathbf{x}; \sigma_{\mathbf{X}})} = \frac{P(\mathbf{v}; \sigma_{\mathbf{X}})}{P_{\sigma_{\mathbf{x}}}(\mathbf{x})} \tag{27}$$

$$= \frac{\sum_{\mathbf{u}} \left( P(\mathbf{u}) \prod_{V_i \in \mathbf{V} \setminus \mathbf{X}} P(v_i \mid \mathbf{pa}_i, \mathbf{u}_i).P_{\sigma_{\mathbf{x}}}(\mathbf{x}) \right)}{P_{\sigma_{\mathbf{x}}}(\mathbf{x})} \quad \text{Truncated factorization product}$$
$$\tag{28}$$

$$= \sum_{\mathbf{u}} P(\mathbf{u}) \prod_{V_i \in \mathbf{V} \setminus \mathbf{X}} P(v_i \mid \mathbf{pa}_i, \mathbf{u}_i) \tag{29}$$

$$= P^{\mathcal{M}}(\mathbf{v}; do(\mathbf{x})) \quad \text{Truncated factorization product}$$
$$\tag{30}$$

Eq. 28 uses the fact that each sub-group assigned $(X_{\sigma_{\mathbf{x}}} = \mathbf{x})$, by independence, has the same frequency of latent features $P(\mathbf{u})$. ∎

**Lemma C.3** (Interventional sample). *An agent $\mathbb{C}$ can draw an i.i.d sample distributed according to the $\mathcal{L}_2$ query $P(\mathbf{V}; do(\mathbf{x}))$ associated with an SCM $\mathcal{M}$, by the following actions:*

*i.* $\text{SELECT}^{(i)}$

*ii.* $\text{RAND}(\mathbf{X})^{(i)}$

*iii. If* $\text{RAND}(\mathbf{X})^{(i)} = \mathbf{x}$, *then* $\text{READ}(\mathbf{V})^{(i)} = \mathbf{v} \sim P(\mathbf{V}; do(\mathbf{x}))$, *else repeat i-iii.*

*Given $N_{\mathbf{x}}$ i.i.d samples, the consistent unbiased estimate of Eq. 24 is given by*

$$\hat{P}(\mathbf{v}; do(\mathbf{x})) =$$
$$\underbrace{\frac{1}{N_{\mathbf{x}}} \sum_{i}}_{\text{②}} \underbrace{\mathbb{1}[\text{READ}(\mathbf{V})^{(i)} = \mathbf{v}, \text{RAND}(\mathbf{X})^{(i)} = \mathbf{x}]}_{\text{①}}, \tag{31}$$

*Proof.* The proof steps are similar to the ones used for the Observational i.i.d sample case. Note that Remark D.3 still hold since even though the agent is conditioning on the value randomly assigned to a particular unit $i$, this value is independent of the unit's latent features $\mathbf{U}^{(i)}$. ∎

