# OpenReview forum: "Counterfactual Realizability"
_ICLR.cc/2025/Conference — ICLR 2025 Spotlight_

### Official Review · Reviewer_4AQU · 2024-11-02

**Soundness:** 3
**Presentation:** 2
**Contribution:** 3
**Rating:** 6
**Confidence:** 4

**Summary:**

Inspired by (Bareinboim et al., 2015), the authors introduced a new action of "counterfactual randomization (ctf-randomization)" in structural causal models. One of the examples of how such action can be applied is given in Definition 2 (Counterfactual mediator). In particular, for some variable $X$ and its children $Y$, $W$ is the counterfactual mediator of $X$ if the mechanism generating $Y$ can retrieve the value of $X$ from $W$. The authors studied under what conditions on the causal structure and set of available actions (which might include counterfactual randomization), it is feasible to generate i.i.d samples from a counterfactual distribution. This study assumes that each unit undergoes each causal mechanism only once (Assumption 3.1). The authors provide graphical conditions along with an algorithm (CTF-REALIZE) to check the feasibility of sampling (which they call this problem "realizability"). Moreover, they characterized graphical conditions for the case that it is possible to perform ctf-randomization for each child of
each variable, and as a corollary, they obtained the case of "fundamental problem of causal inference". They explored the applications of counterfactual realizability in causal fair analysis and causal bandits. In particular, they showed in an example that the interventional fairness metric fails to preserve disparities
in outcomes across groups while a counterfactual one approach works in it. Moreover, in a specific causal bandit setup, for a specific "MAB template", they showed that the proposed $\mathcal{L}_3$-based strategy is optimal.

**Strengths:**

Originality/Quality:
The definition of ctf-randomization and the algorithm for checking the counterfactual realizability are new in the field of causality. Moreover, the authors discussed the applications of $\mathcal{L}_3$-based methods in fairness analysis and bandit problems which might be interesting for people in other fields.

Clarity: The writing should be improved as some of the parts need more explanations.

Significance: Although the concept of counterfactual realizability is new in the field, it is not completely clear to what extent such ctf-randomization can be implemented in reality.

**Weaknesses:**

The content of the paper is fairly dense and the authors did not have enough space to give a sketch of the proofs or even CTF-REALIZE algorithm is not described very well in the paper (For instance, it is not clear why there is a rejection sampling in line 18 of Algorithm 1). This also applies to the results in Section 4. Although the authors showed in some specific scenarios, $\mathcal{L}_3$-based methods perform better than $\mathcal{L}_2$-based methods, there is no intuition given in the paper why this is the case. One might wonder if the causal graphs or the causal mechanism are meticulously engineered to show the advantage of counterfactual reasoning.

**Questions:**

1. For what real applications, is it possible to implement ctf-randomization? To my understanding, this might be only possible for some output $Y$, where it is generated by a computer program. Therefore, it is possible to generate the output several times with different inputs with the same randomization.

2. In the definition of counterfactual mediator, it is assumed that the value of $X$ can be retrieved from $W$ in the causal mechanism that generates $Y$. Does it mean that there is no exogenous noise in $W$ and it is generated deterministicly from $X$?

3. It is good to provide some intuitions on the advantage of using $\mathcal{L}_3$-based methods that are discussed in Section 4.

4. Please provide a clear description of  Algorithm 1 and COMPATIBLE subroutine either in the main body or the appendix. This should start with a general idea of the algorithm or subroutine and then describe the algorithm in detail. In the current version, it is hard to verify the main results for Algorithm 1.

---

> ### Author Response · Authors · 2024-11-20
>
> We thank the reviewer for the great suggestions and questions! We also appreciate the acknowledgment that the result is novel and important to multiple applications. We address your questions below.
>
> (Q1) **Applications**: we appreciate the opportunity to clarify a slight misunderstanding here.  An output $Y$ cannot be generated multiple times for the *same* unit under different randomized values (as formally highlighted by Cor. 3.8). Rather, *ctf-rand( )* can be used to measure the $Y$ for *different* units, each generated under a different randomized condition. In practice, $Y$ *does not* need to be a digital mechanism. In Example 3, $Y$ indicates a user successfully staying within her time limit (each day is a unit). In Example 2, $Y, Z$ could also be independent human evaluators who separately receive hardcopy CVs (each CV is a unit that can be ctf-randomized with different names at the top). Indeed, the field experiment cited in Example 2 involved real-world job applications [1]. The applicability of this technique is quite broad. Still, as noted in Sec. 5, most systems and interactions *are* being digitized, including customer service, financial transactions, remote working, and recruiting, especially with LLMs entering the ecosystem. This topic is only starting to be recognized, and we expect it to become increasingly relevant and offer other possibilities for new types of experimental design that allow one to access highly informative counterfactual information. Formally, the main condition that allows/precludes one from performing this procedure in such settings is based on the construct we call “ctf-mediator”, as elaborated in the next bullet.
>
> (Q2) **Mediator condition**: A ctf-mediator $W$ can receive an exogenous noise variable, but the *inverse* from $W → X$ needs to be deterministic. This means that the domain of $W$ divides into equivalence classes of values that can each be mapped back to a unique $x$ (explained in lines [1724-1727]). This is quite intuitive and aligns with real-life situations. For instance, in Example 2, a race $X=x$ could be associated with many stereotypical names {$\{W=w\}$}, but for each stereotypical name that is chosen for the experiment (e.g. “Nguyen”), it maps back to a unique race (“Asian”). *ctf-rand( )* requires that the outcome variable perceives $X$ via some interface channel that can be manipulated, not necessarily digitally. E.g., the hardcopy CV example earlier, or a market-research study where a participant could be made to perceive different signals by altering the physical study environment.
>
> (Q3) **Experiments**: for Example 2, we describe the intuition for the results in Sec. E.2.2 using a worked numerical SCM where the L2 metric suggests zero discrimination, but the L3 metric is actually non-zero. To ensure we didn’t cherry-pick the numbers, we ran 1,000 optimizations with random seed, which revealed that a divergence between the L2 and L3 metrics occurred in most cases (Fig. 5). This makes sense because L2 is just less informative about the underlying mechanics of the system, and our ctf-randomization procedure is needed to probe L3. For Example 3, we provide a detailed interpretation of the latent variables and mechanisms in lines [2160-2175] to ensure the graph is not arbitrary. The intuition for why L3 performs better is because with each additional *ctf-rand( )*, we are getting more information about the latent variables $U_1, U_2, U_3$ that affect $Y$. Interestingly, we can even use a descendant of $X$, $D$ to obtain information about $Y$ via the confounder $U_3$. This insight is not present in any previous work, to the best of our knowledge!!
>
> (Q4) **Algo walk-through**: We provide step-by-step examples for the algorithm in Appendix B.4. **[NEW]** We have also added a detailed walk-through for the algorithm and proof strategy in Appendix B.2. To summarise: for each variable $V$ in the input graph, we check what are the necessary and sufficient interventions (or lack of interventions) one needs to perform w.r.t each potential outcome $W_{\mathbf{t}}$ in the input query $\mathbf{W}_\star$. This is what the inner loops are doing -- tracking all the required conditions in topological order. If there is no conflict across these conditions collectively, and if the feasible action set contains the necessary actions, the query is realizable. Otherwise not.
>
> We are happy to explain any aspect in more detail! We also explain in Appendix B.2 that rejection sampling was used to keep the presentation clean instead of overlaying concentration guarantees, etc., which distracts from the main point.
>
> We appreciate the meaningful discussion and would be happy to provide further clarification.
>
> Thank you!
>
> **References**:
>
> [1] Bertrand & Mullainathan. 2003. Are Emily and Greg more employable than Lakisha
> and Jamal? A field experiment on labor market discrimination

---

> ### Author Response · Authors · 2024-11-25
>
> Dear reviewer 4AQU,
>
> We were wondering if you had any questions or feedback regarding the algorithm’s walk-through and proof strategy, and the scope of applications of our method.
>
> We are keen to submit the best version of our paper, and would gladly incorporate further suggestions.
>
> Thank you!
>
> Authors of submission 12927

---

> > ### Comment · Reviewer_4AQU · 2024-11-26
> >
> > I appreciate the authors' detailed response. Regarding the additional explanation in Section B.2, I believe further clarification is necessary. The current explanation offers a general summary of the algorithm, but it would be helpful to explicitly indicate which lines of the algorithm correspond to specific parts of the necessary and sufficient conditions. For example, in COMPATIBLE, step ii.b, what exactly is meant by "minimal intervention"? Why is it required, and how is it determined?
> >
> > While I find the results in the paper to be novel, the content is dense (48 pages), making it more suitable for a journal than an ML venue with limited review time. Due to time constraints, I was unable to thoroughly verify all details, and some sections remain unclear to me. I recommend adding more detailed explanations and submitting this work to a journal, which would provide a more appropriate platform for reviewing this work.

---

> ### Author Response · Authors · 2024-11-27
>
> We greatly appreciate the reviewer’s effort in providing feedback, and do sympathize with their constraints during the review cycle. We sincerely hope to use the extended discussion period to resolve any gaps in understanding the main result.
>
> We hasten to reassure the reviewer that all appendix material is *not* needed to understand the main results! As we emphasized throughout the appendices (e.g., at the start of Appendices D and E, lines [1677], [2003], [202]), most of the appendix is targeted at the interested reader: detailed case studies for some compelling causal queries (Appendix B.4), pre-empting FAQs (Appendix D), and discussing novel aspects of experiments (Appendix E), which are common at venues like ICLR.
>
> We believe these discussions would improve the impact of our work, since a diverse audience might be curious about various aspects of the results. For instance, industry practitioners at ICLR may want to learn how to actually implement this new experiment-design technique, causality researchers may be interested in the nuances of the causal hierarchy, and applied ML researchers might be interested in developing new methods in fairness or RL. However, the self-contained results in Sec. 3 can be understood without this further elaboration.
>
> Focusing on our main results in Sec. 3, and to address the feedback that it is dense, we tried to provide an increasingly refined view of the results as follows:
>
>  (1) An intuitive example illustrating the concept: lines [263-300] + Fig. 3
>
>  (2) **[UPDATED]**  A general summary of the algorithm and proof-sketch: lines [301-306, 790-795]
>
>  (3) **[UPDATED]** A detailed walkthrough of the algorithm and proof-sketch: [796-859]
>
>  (4) Full proof of the algorithm and theorems: Appendix C
>
>  (5) Multiple examples (query + graphs) of applying the algorithm: Appendix B.4
>
>
> We believe this strikes a reasonable balance between friendliness/legibility, and also depth, for the readers who may be interested in the details. The audience at ICLR is broad, so our strategy was to provide this stratified view of our results. Still, we are certainly open to improving and reorganizing some parts of the paper, if the reviewers think this would be suitable, since all the content is already there!
>
> Specifically regarding Step ii.b., we have **updated Sec. B.2** with a **detailed** explanation of the necessary and sufficient steps and the minimal intervention needed (changes in blue). The induction argument of why it is enough to check conflicts in topological order is given in the full proof.
>
> Besides step ii.b, are there any details not clear?
>
> We remain fully committed to resolving all gaps in understanding!
>
> Thank you for your engagement!

---

> > ### Comment · Reviewer_4AQU · 2024-11-28
> >
> > I thank the authors for their response. I have a general understanding of the current work, but as reviewers, it is our responsibility to verify the correctness of the claims made in the paper. In my view, the core contribution of the paper lies in the soundness and completeness of CTF-REALIZE, as presented in Theorem 3.5. However, the submitted version lacks an adequate explanation of the COMPATIBLE procedure, which is used as a subroutine in CTF-REALIZE. The only reference to it is the pseudocode in Appendix B.1, having undefined terms like "label" and "Natural" in the submitted version. Without knowing how CTF-REALIZE works, it is impossible to verify the results in Theorem 3.5.
> >
> > The authors have been adding further explanations during the discussion period, which is helpful. I will try to review the proofs during the extended discussion time, but I believe the original submission should have included all the necessary details to ensure clarity and completeness.

---

> > > ### Author Response · Authors · 2024-11-30
> > >
> > > Dear reviewer 4AQU, thank you for the rigor in checking the technical details of our results - we agree that this is important, and do appreciate the meticulous review!
> > >
> > > We hope that the walk-through of the algorithm (Sec. B.2 of the current version) clarifies all the doubts raised: why rejection sampling [iv], necessary and sufficient conditions and minimal actions [ii.b.1 - ii.c.4], and what the tags in the sub-routine mean [ii.b.2, ii.c.3].

---

> ### Author Response · Authors · 2024-12-02
>
> Dear reviewer 4AQU, as the discussion period is drawing to a close, we would like to check if there are any further questions that we can address in our final response.
>
> Thank you!
>
> **EDIT**: We wish to highlight **(for the benefit of the Meta-Reviewer)** that we have fully addressed all the clarifications requested by reviewer 4AQU in our submission.
>
> Some of these clarifications are not material to the correctness of our main result (e.g., why we used rejection sampling), and some of these clarifications were already present in the original submission’s main body (e.g. what are the necessary and sufficient actions was described in the original submission in Sec. 3 lines [299-306]), or discussed in the proof.
>
> Still, the reviewer’s suggestion of providing a more detailed walk-through of the algorithm was a good one, and we have incorporated this into the final version of our submission in Appendix B.2.

---

### Official Review · Reviewer_doTz · 2024-11-04

**Soundness:** 3
**Presentation:** 3
**Contribution:** 4
**Rating:** 10
**Confidence:** 4

**Summary:**

The authors provide necessary and sufficient conditions for a ground truth counterfactual quantity of an SCM to be sampled directly (''realized'') through a ''counterfactual experiment''. Such an experiment involves manipulating the original SCM through adding mediators through some direct causal effects---a process the authors motivate with intuitive real-world examples.

**Strengths:**

The authors' contributions are original and significant. Their insights are laid out clearly and limitations and discussion of future work are provided.

The authors show the advantages of counterfactual experiments with intuitive examples. The authors show that counterfactual fairness measures that are not identified through interventions can be realized (and thus identified) through counterfactual experiments. Furthermore, the authors show that bandit policies which allow for counterfactual mediators dominate policies from the classical causal bandits setting.

**Weaknesses:**

I do have few substantial weaknesses to discuss. In the Questions section I voice a slight disagreement about the interpretation of ''physically possible'' experiments. Below are some minor comments.

(Lines 105--107) Presumably W_* is a tuple of potential outcomes and w is a vector over the support of W_*. These should be defined formally to avoid confusion. I suspect there is a typo on line 107 when defining P^M---perhaps taking the product over t = 1 to |W_*| would clear this up.

(Lines 120--121) Read (V)^(i) does not ''measure'' the effect of f_V on V, these mechanisms are in general not identified. This should be reworded---perhaps the physical actions should be defined formally.

(Figure 2) Given the discussion in Example 1, it may be more natural to include latent confounding between X and Z in the DAGs.

**Questions:**

I would argue there do exist physically viable experiments that break the authors’ proposed fundamental constraint on experimentation. For instance, in the authors’ example of an artificial agent’s decision to issue a speeding ticket from video footage based off the color of a car, both the doctored footage of a yellow car and the original footage of a red car could be fed to multiple copies of the same artificial agent. Perhaps the assumption is more properly termed a ''constraint on experimentation without cloning’’?

---

> ### Author Response · Authors · 2024-11-20
>
> We thank doTz for their meticulous review and feedback. We are heartened by your acknowledgment of our contribution and its implications.
>
> **On the fundamental constraint of experimentation (FCE):**
>
> This is an excellent point and gets to the motivation of the FCE assumption. There are two possible settings one could discuss, namely:  1. The full model of the environment (SCM) and the latent variables $\mathbf{U}$ per unit are known; and 2. The SCM and the unit’s identity are unknown.
>
> The first setting involves cases like running simulations, where the experimenter has full access and control over the environment. It could also involve working with an AI agent where you can observe and freeze all model parameters (including noise terms) and re-run inference with different inputs. This seems to be what you describe re: multiple copies. In this setting, we don’t actually need many causal inference tools like identification, and we could just compute values directly from the SCM (in a flavor similar to the semantic evaluation given in Sec. 1.3 in [1]).
>
> In our work, we study the second setting (see also Assumption C.1). In Example 1, this might mean $Y$ is a closed-source AI agent with some temperature setting generating noise terms $U_Y$ in each round. For the same input, different results could be obtained from repeated attempts on the same agent. I.e., we wouldn’t be able to freeze the random seed and evaluate the outcome under doctored yellow-color and red-color. For a single round, we are limited to just one set of inputs $W=yellow$. Repeating mechanism $Y$ again with $W=red$ means the unknown $U_Y$ term will have changed. In this case, this does not match the counterfactual definition, since the quantities are not being scoped relative to the same unit.
>
> Importantly, this second setting realistically represents how we will interact with AI agents and many other interactions in the real world when the underlying mechanisms (or human behavior) are unknown. For instance, in Example 1, the experimenter is an auditor making API calls to a closed-source proprietary system they cannot control. Also note: even the so-called fundamental problem of causal inference (Cor. 3.8 and [2]) wouldn’t arise under your setting of freezing a model.
>
> **Minor comments:**
>
> - Line [111]: Yes, there was a typo! $\mathbf{W}_*$ is a random vector containing multiple potential outcomes, and $\mathbf{w}$ is a specific vector value drawn from the support of the former. The external summation is over units $\mathbf{U}=\mathbf{u}$ that would deterministically induce the random vector to take the value $\mathbf{w}$. The inner product of indicators should check whether each potential outcome $W_t$ in the random vector takes the required value in $\mathbf{w}$ - this is corrected now!
>
> - Def. 2.1: The way we think about it, $f_V$ is the mechanism which, you are right, is not observed in general. It produces some observable feature $V$ for the unit. And READ is the action of recording this observed value. We have reworded slightly to reflect this.
>
> - Confounder in Fig. 2. Also makes sense!
>
> **References**:
>
> [1] Bareinboim et al. 2022. On Pearl’s Hierarchy and the foundations of causal inference
>
> [2] Holland. 1986. Statistics and Causal Inference

---

> > ### Comment · Reviewer_doTz · 2024-11-24
> >
> > Many thanks to the authors for their response and clarifying my questions with respect to their proposed fundamental constraint.
> >
> > To my mind, the authors present an important contribution to the foundations of causal inference, with clear implications for experimentation and policy. The authors' examples of optimal 'counterfactual policies' in a causal bandits setting and newly-realizable counterfactual fairness measures make this clear.
> >
> > I have adjusted my review to recommend that this work to be highlighted at ICLR.

---

### Official Review · Reviewer_4NTd · 2024-11-06

**Soundness:** 3
**Presentation:** 2
**Contribution:** 2
**Rating:** 6
**Confidence:** 2

**Summary:**

The authors tackle the complex space of sampling from counterfactuals.

Specifically, they propose a "formal definition of realizability, the ability to draw samples from a distribution, and then developing a complete algorithm to determine whether an arbitrary counterfactual distribution is realizable given fundamental physical constraints, such as the inability to go back in time and subject the same unit to a different experimental condition"

Both causal fairness and CRL are used as illustrative examples, suggesting that a counterfactual strategy dominates strategies based only on interventional and observational data.

**Strengths:**

The paper tackles a relevant problem, and covers all key bases to provide its key contribution.

Sampling from counterfactual distributions is naturally hard, and developing methods and tools to do so is important, as suggested in the paper.

**Weaknesses:**

I am not an expert in this field, so take it with a grain of salt:

As written by the authors "representing counterfactual distri- butions, is believed to be inaccessible almost by definition."

References are provided for e.g. the 'counterfactual randomisation procedure', i.e. attempts at sampling from CFs, but there is not much discussion about why it is 'believed' why CFs are inaccessible by definition.

"inaccessible almost by definition" is not precise enough to be useful for a reader, "almost" needs to be refined and explicated, ideally both via intuition and formally, even just briefly.

In the conclusion, this rigour is missing, too, when the authors state: "Countering prevalent belief". A reader needs references to be able to follow the thoughts and verify by themselves, that indeed, the venture of CFs sampling is hard (or impossible?)

If CFs are not in fact ""inaccessible almost by definition"", then this paper of course would constitute a huge contribution to the field of causal inference in which case this should be highlighted more.

**Questions:**

- [ ] Bounding in mentioned but not referenced? The word "bounding" appears twice in the text, but at first mention is not referenced. Bounding is not a well known method, and without further definition will not be useful to the reader unless at least minimal referenced (and not just in the discussions section)
- [ ] 535 Countering prevalent belief
    - [ ] References pls, readers need to be able to verify claims independently
- [ ] 168 fig 2a does not exist, do you mean top?
- [ ] Assumption 3.1: I worry that this statement is too vague for a reader to agree with. Can you provide more justification for why this is reasonable to hold in practice? If not, provide justification for why a practitioner should entertain this method to provide additional analysis instead.
- Can you define "causal machine learning"? I am aware that this is an emerging term, but in the context of this paper it is unclear to me what part of it the authors are trying to address exactly.

Typos:
- [ ] 507 typo "cotribution"
- [ ] 175 mechanism OF





- [ ] 258 what is an intuition pump?
- [ ] 431 appreciate limitations
    - [ ] What about limitations in general though?
- [ ] 507 typo
- [ ] 507: what bounding do you mean here? References?

---

> ### Author Response · Authors · 2024-11-20
>
> We thank the reviewer for the insightful comments. We are heartened by your interest in this topic! We believe this is an important result for the field of causal inference, and are optimistic that the above concerns can be successfully addressed.
>
> Your main questions are (pls correct us if we missed anything!) : what is/are
>
> (a) some intuitive and formal reasons why counterfactuals are believed to be non-realizable by definition?
>
> (b) some references pointing to this conventional belief?
>
> (c) bounding, and related references?; and
>
> (d) some topics we mean by “causal machine learning”, so that you can place our work in context?
>
> We appreciate the candor that you are not as familiar with the subject.
>
> To respond:
>
> (a) A counterfactual distribution is some $P(W_*)$ where $W_*$ is a tuple of potential responses from different regimes (we explain this in Preliminaries; also see Fig. 1.13(iii) in [1]). E.g., suppose $X$ is a drug dosage and $Y$ is a health outcome. The joint distribution $P(Y_x, Y_{x’}, X)$ means we are reasoning about the probability that $Y$ would have taken some value $y$ had we done $do(x)$ for a patient in a randomized trial, *and* $Y$ would have taken some value $y’$ had we instead done $do(x’)$ for the *same* patient, *and* the patient would naturally have been assigned $X=x’’$ in the absence of any randomized intervention. By definition, these scenarios evoke “parallel worlds”. For any given patient, since we can only study her under one experimental condition (world), it is generally believed that such counterfactuals cannot be physically realized.
>
> (b) This is actually a really common belief, sometimes taken as conventional wisdom. We have updated in footnote 1 the following quotes from influential researchers: *“By definition, one can never observe [counterfactuals], nor assess empirically the validity of any modeling assumptions made about them”* [2], and also the quote *“...The problem with counterfactuals like [$P(Y_x \mid x’)$] is [that] … we simply cannot perform an experiment where the same person is both given and not given treatment”* [3]. But as we explicate in Example B.1, $P(Y_x \mid x’)$ indeed is experimentally realizable, by measuring $Y_x$ and $X$ in the same world (i.e., we can “realize” this without literally entering parallel worlds). Still, the belief has persisted that *“counterfactual judgments remain hypothetical, subjective, untestable, unfalsifiable”* [4], due to the understanding that parallel worlds are inaccessible. This matters because agents who don’t leverage this capability (believing it impossible based on this limited understanding) can suffer badly, as Examples 2 and 3 illustrate. Note: the other reviewers acknowledge that the formal results are novel.
>
> On a related note, we respectfully disagree re: Assumption 3.1. We feel it is evident that in most real-world cases a physical process, once performed, cannot be reversed without changing underlying features, with a justification sketched in [227-236]. Relaxing this assumption may be less realistic for practitioners.
>
> (c) The main reference is [5] which is now clarified in Sec. 5 lines [505-508]! Bounding is a task within the identification setting. Namely: using causal assumptions such as a causal graph, we can show that a causal query is bounded within some range computable from the available data. If the query is “identifiable” from the input data, the range collapses to an exact value. As discussed in Sec. 5, it stands to reason that if we can get more kinds of data from richer experiments, the bounds can only tighten further.
>
> (d) We mean areas like causal fairness analysis, causal RL, causal generative modeling, and others that leverage the tools of causal inference to improve the performance, trustworthiness, and scope of ML methods. The excellent survey in [6] covers some of these but is slightly dated (it doesn’t cover some recent developments). We highlighted at least three areas in our examples where counterfactual realizability is directly relevant: explainability (mediation), fairness analysis, and reinforcement learning.
>
> Please let us know if this makes sense and if you have further feedback!
> We are fully committed to resolving all pending issues.
>
> **References**:
>
> [1] Bareinboim et al. 2022. On Pearl’s Hierarchy and the foundations of causal inference
>
> [2] Dawid. 2000. Causal Inference Without Counterfactuals (with Comments and Rejoinder)
>
> [3] Shpitser & Pearl. 2007. What Counterfactuals Can Be Tested
>
> [4] https://www.inference.vc/causal-inference-3-counterfactuals/
>
> [5] Zhang et a. 2022. Partial counterfactual identification from observational and experimental data
>
> [6] Kaddour et al. 2022. Causal Machine Learning: A Survey and Open Problems

---

> > ### Author Response · Authors · 2024-11-20
> > **Typos**
> >
> > PS: we really appreciate the typos!!
> >
> > - [175], now [180], is actually correct. $f_Y$ is the mechanism generating $Y$
> >
> > - [258], now [264]: intuition pump is just a colloquial term for some thought-experiment or example designed to build intuition
> >
> > - The rest are corrected.

---

> ### Comment · Reviewer_4NTd · 2024-11-20
> **Thank you for the clarifications**
>
> And I deeply appreciate you allowing me to learn from you and update my conventional belief.
>
> All questions are addressed and I have adjusted my score accordingly.

---

### Official Review · Reviewer_jnLW · 2024-11-08

**Soundness:** 3
**Presentation:** 3
**Contribution:** 3
**Rating:** 8
**Confidence:** 4

**Summary:**

This paper considers a simple procedure that makes it possible to sample certain counterfactual distributions (counterfactual realizability). The key observation is that the perception of a variable to its children can be intervened upon in some cases. This procedure is called counterfactual randomization. It also proposes an algorithm that takes in a candidate counterfactual distirbution and a graph, and outputs a sample from the counterfactual query if and only if it is possible to sample from the input candidate. Finally, a graphical characterization for counterfactual realizability is given.

**Strengths:**

As the paper points out, the counterfactual randomziation procedure in itself has been alluded to in previous work. But, the main contribution of this paper is a characterization of when a counterfactual distribution can be 'realized'. I liked that the paper operates under this structure for sampling distributions at different levels of the causal hierarchy by the notion of actions. Since counterfactuals are commonplace in many applications and a major hurdle is their non-identifiabilty, this work is certainly significant. The paper is well written and padded with many examples to illustrate the applicability and importance of the procedure.

**Weaknesses:**

As the paper also mentions, I believe the idea for intervening on perceptions was known before and makes multiple informal appearances. A few other references are (S. Geneletti, A.P. David, Defining and identifying the effect of treatment on the treated, 2010). Also, the notion of SWIGs (Richardson and Robins, 2013) is quite similar. I believe a more thorough comparison with previous work is needed to make the paper more complete.

**Questions:**

1) One interpretation of the counterfactual randomization procedure is that the counterfactual distribution is just an interventional distribution where the intervention is performed on the perception. Given that a few counterfactual distributions are realizable as defined, how does this affect the notion of 'causal hierarchy'?  Some discussion on this would certainly strengthen the paper.
2) The main body and the proofs often use the notion of a probability measure of belief of an exogenous agent. I thought that this was ill-defined. Perhaps the authors could make Remark C.7 an axiom that defines this measure?

---

> ### Author Response · Authors · 2024-11-20
>
> We thank jnLW for the valuable feedback and expertise, and appreciate the opportunity to discuss some of these nuanced aspects of our work!
>
> > A few other references are (Geneletti & David, 2010) and the notion of SWIGs (Richardson & Robins, 2013)
>
> Indeed both are very relevant papers! We do cite the latter in Sec. 5 to explain the difference vs. our work. We now added the former in Sec. 5 line [502] as well.
>
> Geneletti & David cast the unit’s natural decision as a *separate* variable, say $T$, and use identification arguments to compute ETT. This framing does not lend itself to a causal RL algorithm which optimizes $\mathbb{E}[Y_x \mid X=x’]$. In the latter, it is very clear when exploring the arms of the bandit problem that when $x = x’$, you can hot-start the learning using the counterfactual relationship between the variables (e.g. Equation 41 in Appendix E.3.1). If you frame it as $\mathbb{E}[Y_x \mid T=t]$, this would be the “mere” contextual bandit problem we discuss in lines [486-490]. The second reference does not face this issue, and does cast $T$ as the natural decision variable $X$.
>
> Still, this discussion highlights our contribution perfectly. As explained in lines [210-212], *ctf-rand( )* in our framing differs from Fisherian *rand( )* in two ways: (1) the natural value of $X$ is leveraged, and (2) where possible, counterfactual mediators allow for multiple simultaneous randomizations of $X$, each affecting different subsets of children.
>
> Both Geneletti & David and Richardson & Robins acknowledge (1) but not (2). (2) is not possible in their formalism. Consequently, their approach wouldn’t discover the *“optimal L3 strategy”* we discovered in Example 3 in Table 1 and Fig. 6(c-d; purple plot), which optimizes $\mathbb{E}[Y_x \mid x’, d_{x’’}]$. They would be limited to the *“ETT baseline strategy”* that we show in Table 1 and Fig. 6(c-d; red plot), which optimizes $\mathbb{E}[Y_x \mid x’]$. Further, the approach in both references can’t be applied in Examples 1 and 2 either, since these examples require path-specific randomizations. Still, we added the reference as mentioned.
>
> > (Q1) Interpreting counterfactual randomization as an L2 intervention on the perception.
>
> This is a great point, and touches on the core issue. We want to clearly establish that the difference between L2 and L3 is not about realizable vs. non-realizable. Rather, L3 is about sampling joint potential outcomes from different regimes.
>
> Yes, we could technically construe ctf-randomization as L2 randomization on perception. But this misses the counterfactual connection between perceived treatment and different potential outcomes via structural constraints. Philosophically, this really is a different kind of randomization procedure we are formalizing.
>
> In a similar vein, we could technically see an L2 distribution like $P(Y; do(x))$ as “just” an L1 distribution, albeit in a different environment that has undergone a “distribution shift”. But this misses the whole connection between $P(Y; do(x))$ and $P(Y \mid x)$ via structural constraints in the SCM. Philosophically, this justified the development of a new L2 semantics that captures that connection.
>
> > (Q2) Perhaps the authors could make Remark C.7 an axiom that defines this measure?
>
> Good suggestion, thank you! We have added lines [1206-1214] in Appendix C.1 (Assumptions), to clarify the probability measure of the agent’s belief. Let us know if that makes sense.
>
> We welcome further suggestions for improving the paper, and appreciate the thought-provoking engagement!

---

> > ### Comment · Reviewer_jnLW · 2024-11-26
> >
> > Sorry for my delayed follow-up.
> >
> > "But this misses the counterfactual connection between perceived treatment and different potential outcomes via structural constraints. Philosophically, this really is a different kind of randomization procedure we are formalizing." - I don't see the counterfactual connection you point out that is missing if I treat your ctf-rand procedure as an intervention on perception of a variable. I am glad you think this "touches on the core issue" since it perhaps necessitates a more thorough explanation with an example in the paper? Because your point later about viewing an interventional distribution as an observational distribution is not exactly the same. The distribution shift is not brought upon by any "action" (using your formalism) whereas ctf-rand is rand on the perception (which I understand your definition doesn't allow since it randomizes only modeled decision variables) which is more plausible?

---

> > > ### Author Response · Authors · 2024-11-30
> > >
> > > We greatly appreciate your engagement! This discussion is useful to help us frame and communicate our results more effectively.
> > >
> > > **Background:**
> > >
> > > Building on Pearl’s original work [1], interventions and counterfactuals are qualitatively different, with the formal distinction between the two later developed in [2]. Interventions deal with changes to the current world, e.g. $P(Y_{X=x} = y)$, where $X=x$ is the intervention. Counterfactuals deal with alternative scenarios, and a basic example is the Effect of Treatment on the Treated, $P(Y_{X=x} | X = x’)$, which was first introduced in [3] to analyze the effect of a job training program on those who voluntarily enrolled. One hypothesis was that the program was not as effective as the data suggested, because the candidates who voluntarily enrol tend to be more qualified to begin with, and might have achieved similar outcomes even without the training. In words, $P(Y_{X=x} = y | X = x’)$ reasons about the odds that $Y=y$ had $X=x$, for those who naturally chose $X=x’$. This clearly shows the contrary-to-fact nature of the statement since $x’ \neq x$. Another counterfactual quantity is the probability of sufficiency, $P(Y_{X=x} = y | X = x’, Y = y’)$. In the factual world where $X = x’, Y = y’$, one might wonder whether $Y = y$ would have occurred had $X = x$  [1, Ch. 9].
> > >
> > > Interestingly enough, interventions are immediately linkable to randomized experiments, while counterfactual quantities were believed to be physically unrealizable, and only testable through symbolic identification. However, [4] noted that under certain conditions, there is a different type of randomization that allows one to sample directly from the specific counterfactual distribution $P(Y_{X=x} = y | X = x’)$. Our work substantially refines this understanding and provides more precise, general conditions for when counterfactual randomization is possible. Given that this procedure is a deviation from Fisherian randomization and allows measurements of counterfactual quantities, we feel that calling it *counterfactual randomization* would allow one to distinguish the subtleties and additional care required in this setting. One important requirement of this procedure, qualitatively different from Fisherian randomization, is that it requires some assumptions (graphical knowledge, presence of a counterfactual mediator etc).
> > >
> > > **Your question:**
> > >
> > >  Consider a decision variable $X$, outcome $Y$, and perception variable $W$. Note that by Def. D.2 of a ctf-mediator and Lemma D.4, randomizing $W$ is akin to measuring $Y$ *had $X$ been fixed as $x$* - a counterfactual statement. If we understand your question correctly, you are asking if we can w.l.o.g describe *ctf-rand( )* as simply the randomization of $W$, instead of a counterfactual randomization of $X$. We could - but then we would need to spell out additional statements each time that $W$ has certain properties, and is only a proxy by which the $X$ is perceived by $Y$ (Lemma D.4) etc.
> > >
> > > - In Example 1, you could frame it as a randomization of RGB values, or you could frame it as randomization of car color ($X$) *as perceived* by the model ($Y$). Both are valid descriptions, and indeed the former is how we actually implement it. But the reason we are interested in the *ctf-rand( )* procedure to begin with is because it lets us counterfactually fix $X$ as the input to mechanism $Y$. Def. 2.3 of counterfactual randomization abstracts out the intermediate steps (Lemma D.4) needed to reason that RGB value is a proxy for color. We believe that, when dealing with counterfactual mediators (e.g., color -- RGB) and not just regular mediators (e.g., disease -- symptoms), this connection is more intuitively expressed as “counterfactual” randomization of $X$ itself.
> > > - In Example 3, the $W$ would trivially be the log-in action by which the variable $D$ (type of ads) perceives the natural log-in method $X$. It is semantically clearer to omit the $W$ and frame this as  *ctf-rand*$(X → D)$ illustrated in 6(b), and thereby to deploy an RL algorithm that directly optimizes $E[Y_x \mid x’, d_{x’’}]$. Adding a perception variable $W$ into the picture, and reasoning about the connection of $W$ with $X$ in the algorithm, would be redundant without adding much insight.
> > >
> > > As you point out, randomization of perception has been mentioned earlier. And it has been used to reason about counterfactual modifications of $X$ itself, *but typically in an informal/implicit way*. Counterfactual randomization (Def. 2.3) encapsulates formally what is being assumed.
> > >
> > > We discuss this in [192-203], but could elaborate more in Appendix D, if accepted!
> > >
> > > **References**
> > >
> > > [1] Pearl. 2000. Causality
> > >
> > > [2] Bareinboim et al. 2022. On Pearl’s Hierarchy and the foundations of causal inference
> > >
> > > [3] Heckman & Robb Jr. 1985. Alternative Methods for Evaluating the Impact of Interventions
> > >
> > > [4] Bareinboim et al. 2015. Bandits with unobserved confounders: A causal approach

---

### Author Response · Authors · 2024-11-20
**Thank you and feedback incorporated**

We thank the reviewers for their insightful feedback and extremely helpful suggestions. We are appreciative that they recognize our work as *“significant,” “well written,”* and having *“many examples to illustrate the applicability and importance”* (reviewer jnLW); that the contributions are *“original and significant”*, with *“intuitive examples”* and *“insights clearly laid out”* (reviewer doTz); that the main results are *“new in the field,”* and the applications might be relevant in fields like fairness and RL (reviewer 4AQU); and that if the central claim is true that we are correcting a common misperception about counterfactuals, this *“would constitute a huge contribution to the field of causal inference”* (reviewer 4NTd).

**EDIT:** We appreciate that reviewer doTz recognizes our work as an *"important contribution to the foundations of causal inference, with clear implications for experimentation and policy"*. We believe that Fig. 4 in particular is indeed a foundational contribution to the field, summarizing an important result about counterfactuals. We also believe this would be valuable to audience at ICLR, both theoreticians and practitioners - as we highlight in Sec. 4, ignoring this possibility of counterfactual experimentation could lead to serious underperformance or unfairness of ML methods.

**We take note of the critiques**. Prominent among these were:

(a) a request for more background about why counterfactuals are considered hard to evaluate from 4NTd;

(b) clarifications about when ctf-randomization can be conducted from 4AQU and doTz;

(c) request for a more detailed walk-through of the algorithm from 4AQU; and

(d) feedback about framing some remarks and assumptions from jnLW and doTz.

**Our main follow-up is as follows** (detailed points in separate threads with each reviewer):

- Provided a detailed discussion and references to 4NTd, justifying why there is a conventional belief that counterfactuals are hard to evaluate directly and, therefore, why our contributions matter.

- Added in Appendix B.2. a more detailed walk-through of the algorithm, in addition to the description and example already in the main body.

- Provided intuition for why L3 strategies outperform L2 in Examples 2-3 to 4AQU.

- Provided clarifications about real-world applications where the ctf-randomization procedure can be enacted to 4AQU and doTz.

- Corrected typos; edited the Introduction and Def. 2.1; added references.

---

### Meta-Review · Area_Chair_pmGD · 2024-12-22

**Metareview:**

The notion of counterfactual realizability is introduced, where randomizations go along "edges" instead of the vertices themselves. It has relations to the original idea of path-specific effects of Pearl and, as mentioned by the authors, notions of changing the "input" to nodes as in e.g. "perception interventions" as discussed in an example in fairness analysis by Pearl et al. (2016). Related notions also appear in the interpretation of the effect of treatment on the treated (it's interpretation of "effect of the treatment on the treatable", as in Genelleti and Dawid), and also discussed in the paper.

Related ideas however appear in non-counterfactual notions of mediation that could be covered in more detail. That literature goes far beyond the ETT and it's really closely related to what the authors discuss. Please discuss some examples in that direction:

* the "edge interventions" of Shpitser and Tchetgen Tchetgen ("Causal inference with a graphical hierarchy of interventions", Annals of Statistics 2016)
* Geneletti's mediation analysis ("Identifying direct and indirect effects in a non-counterfactual framework", JRSS B 2007)
* Didelez et al.'s mediation analysis ("Direct and indirect effects of sequential treatments", UAI 2006)
* Robins et al.'s separable effects ("An Interventionist Approach to Mediation Analysis", in Probabilistic and Causal Inference: The Works of Judea Pearl, 2022)
* other papers in general about interventionist mediation analysis - it's something also found within the bibliography of the papers above, as well as numerous follow-up results

Some of the details of the algorithms should also be more fleshed out in case of an ICLR publication.

All in all, a relevant contribution of broad interest that can be strengthened by linking to the more mainstream modern mediation analysis literature.

**Additional Comments On Reviewer Discussion:**

An overall productive discussion focusing on motivation and clarity. Reviewer 4AQU was particularly helpful in pointing out ways by which clarity can be improved, and I hope authors follow closely  their advice.

---

### Decision · Program_Chairs · 2025-01-22

Accept (Spotlight)